# Arctic sea ice signatures: L-Band brightness temperature sensitivity comparison using two radiation transfer models

Friedrich Richter[1], Matthias Drusch[1], Lars Kaleschke[3], Nina Maaß[3], Xiangshan Tian-Kunze[3], and Susanne Mecklenburg[2]

[1]European Space Agency, ESA-ESTEC, 2200 AG Noordwijk, the Netherlands
[2]European Space Agency, ESA-ESRIN, Via Galileo Galilei, casella postale 64 - 00044, Frascati, Italy
[3]Institute of Oceanography, University of Hamburg, Bundesstraße 53, 20146 Hamburg, Germany

*Correspondence to:* Friedrich Richter (friedrich.richter@esa.int)

**Abstract.** Sea ice is a crucial component for short-, medium- and long term numerical weather predictions. Most importantly changes of sea ice coverage and areas covered by thin sea ice have a large impact on heat fluxes between the ocean and the atmosphere. L-Band brightness temperatures from ESA's Earth Explorer SMOS (Soil Moisture and Ocean Salinity) have been proven to be a valuable tool to estimate thin sea ice thickness. Potentially, these measurements can be assimilated in forecasting systems to constrain the ice analysis leading to more accurate initial conditions and subsequently more accurate forecasts. As a first step, we use two different radiative transfer models as forward operators to generate top of atmosphere brightness temperatures based on ORAP5 model output for the 2012/2013 winter season. The simulations are then compared against actual SMOS measurements. The results indicate that both models are able to capture the general variability of measured brightness temperatures over sea ice. The simulated brightness temperatures are dominated by sea ice coverage and thickness changes most pronounced in the marginal ice zone where new sea ice is formed. There we observe largest differences of more than 20 Kelvin over sea ice between simulated and observed brightness temperatures. We conclude that the assimilation of SMOS brightness temperatures yield high potential for forecasting models to correct for uncertainties in thin sea ice areas and caution that uncertainties in sea ice fractional coverage may induce large errors.

## 1 Introduction

A valuable application for sea ice observations is the assimilation into forecasting models. The assimilation of fractional sea ice coverage derived by remote-sensing techniques has been proven to be very beneficial and is conducted in various studies (e.g. Stark et al. (2008); Lindsay and Schweiger (2015)). The sea ice age derived from sea ice drift measurements from the National Snow and Ice Data Center (NSDIC) has been used as a proxy for sea ice thickness in the Arctic Sea Ice Volume reanalysis Pan-Arctic Ice Ocean Modeling and Assimilation System (PIOMAS) (Zhang and Rothrock, 2003). However, while assimilating sea ice concentration or age has become quite common, the direct assimilation of Arctic-wide remote-sensing based sea ice thickness is relatively new (Yang et al., 2014; Xie et al., 2016)

For the first time, the European Space Agency Earth Explorer mission SMOS (Soil Moisture and Ocean Salinity) delivers brightness temperature measurements at 1.4 GHz on a global scale (Mecklenburg et al., 2012), which have been used to estimate

sea ice parameters, such as sea ice thickness (e.g. Kaleschke et al. (2012), Tian-Kunze et al. (2014)), sea ice concentration (e.g. Gabarro et al. (2016)) and snow coverage (Maaß et al., 2015). Microwave radiation at 1.4 GHz (L-Band) is especially useful to derive thin sea ice thickness as it is able to penetrate snow and sea ice for more than half a meter and closes the gap to thicker sea ice thickness retrievals of more than 1 meter that use altimetry (Kwok and Cunningham, 2008; Kaleschke et al.,

2010; Ricker et al., 2014; Tilling et al., 2016). This capability is especially important as the Arctic Ocean shifts to a new state, in which older, thicker sea ice is being replaced by younger and thinner ice (Laxon et al., 2013; Meier, 2015).

The L-Band based sea ice thickness retrievals have been validated by comparison to independent data. In preparation for SMOS, Kaleschke et al. (2010) showed a significant agreement between sea ice thickness derived from L-band brightness temperatures and helicopter based electromagnetic (EM) induced sea ice measurements in the Baltic Sea. After the launch of

SMOS in 2009, the sea ice thickness retrieval has been successfully validated with MODIS thermal infrared imagery data in the Kara and Laptev sea (Kaleschke et al., 2012), and the first Arctic-wide maps of thin sea ice thickness were created. To provide the sea ice thickness product, (Kaleschke et al., 2010) applied a incoherent radiative transfer model Menashi et al. (1993), which was further extended with a thermodynamic sea ice model to consider variations of ice temperature and salinity, as well as a statistical sea ice thickness distribution (Tian-Kunze et al., 2014). The retrieved ice thickness correlates with ship-

and airborne observational thickness up to 1.5 m (Kaleschke et al., 2016). The produced ice thickness maps can be used for assimilating sea ice thickness into numerical models.

Ice thickness has been identifies to be an important predictor of Arctic ice extent (Day et al., 2014), and SMOS sea ice thickness has been assimilated into the Massachusetts Institute of Technology general circulation model (MITgcm) with a localized Singular Evolutive Interpolated Kalman (LSEIK) filter (Yang et al., 2014). The assimilation of SMOS sea ice thickness into the

MITgcm model leads to improved ice thickness forecasts, as well as better sea ice concentration forecasts. Furthermore, SMOS sea ice thickness of less than 0.4 m has been assimilated in the TOPAZ Arctic Ocean reanalysis with the result of reduced Root Mean Square Deviations (RMSD) between SMOS and TOPAZ reanalysis sea ice thickness in March and November 2014 (Xie et al., 2016). It was found that the inconsistency between sea ice concentration of the model and the sea ice concentration based on SMOS data is one of the major limitations for a sea ice thickness assimilation.

To improve the accuracy of the sea ice thickness assimilation it is possible also to consider other sea ice parameters, such as the sea ice fractional coverage or ice/snow surface temperature. However, this approach may imply many obstacles, such as different spatial and temporal resolutions from various sensors. A more elegant way is to assimilate brightness temperatures rather than the retrieved sea ice parameters into the model. Brightness temperatures depend on more than one ice parameter, and the advantage of assimilating brightness temperatures into the model is that a wide range of consistent input data is avail-

able from the model. In contrast, independent retrievals force radiative transfer models with input that relies on assumptions, parameterizations and independent auxiliary data, such as climatologies or secondary reanalysis products.

The question arises which radiative transfer model suits best to be used as a forward operator in a brightness temperature assimilation scheme for thin sea ice thickness. So far, simulated brightness temperatures have been validated for idealized typical Arctic conditions (e.g. Maaß et al. (2013), Tian-Kunze et al. (2014)), but have never been compared to L-Band remote

sensing observations on a large scale. Moreover, there is a wide range of different radiative transfer models that can be used

for this purpose. Depending on the required accuracy, the available auxiliary information and the considered wavelength in relation to the expected scatterer sizes, an incoherent one-ice-parameter approximation or a coherent, more realistic multi-parameter model based on the Maxwell equations can be used. However, identifying the optimal model is challenging because observations of all the involved ice parameters in the Arctic, especially together with radiation measurements in the range of

1.4 GHz, are rare and validation is thus difficult.

Here we investigate the Arctic-wide performance of the radiative transfer models of Kaleschke et al. (2010) and Maaß et al. (2013) to simulate brightness temperatures and to identify the most important input parameters for a sea ice thickness application. The first radiative transfer model is the operational model of the SMOS sea ice retrieval ((Kaleschke et al., 2012)), which has been proven to be a simple tool, based on radiative transfer equations that assume only one bulk ice layer and consider

only first-order reflections and refractions at the layer boundaries (e.g. at the water-ice interface). It is computationally very effective. The second radiative transfer model has a more comprehensive representation of the complex structures of sea ice in that it considers multiple sea ice layers and a snow layer on top and takes into account higher-order reflections and refractions at the layer interfaces. In preparation for a brightness temperature assimilation, for both radiation models we concentrate on the input data of the global ocean reanalysis product ORAP5 (Ocean ReAnalysis Pilot 5) produced by the ECMWF (Zuo et al.,

15  2015).

In this study, we evaluate which radiative transfer model to use for assimilating sea ice thickness into the ORAP5 reanalysis by comparing simulated and observed brightness temperatures from the radiative transfer models with ORAP5 input data and SMOS observations, respectively.

## 2 Data and Methods

## 2.1 SMOS brightness temperatures

SMOS is equipped with a passive microwave 2D-interferometer called MIRAS (Microwave Imaging Radiometer with Aperture Synthesis) operating in L-Band at 1.4 GHz (~21 cm). It measures brightness temperatures in full-polarization up to 65° incidence angle every 1.2 seconds (Kerr et al., 2001). The hexagonal snapshots have a swath-width of around 1200 km, which allows a global coverage. Each point on earth is observed at least once every three days with a daily coverage in the polar

regions due to SMOS quasi-circular sun-synchronous orbit at 758 km height.

SMOS snapshots can be influenced by Radio frequency interference (RFI) rooting from radar, TV and radio transmission (Mecklenburg et al., 2012). To account for the most critical disturbances, a RFI filter has been utilized. Brightness temperatures above 300 K identify a snapshot to be RFI-contaminated and are ignored for the brightness temperature product. Values higher than that are not expected to be seen in the Arctic between November and March as the physical maximum of a surface with

temperature at the freezing point would be 273.15 K if the emissivity was 1.

The brightness temperature product is provided at vertical and horizontal polarization. Although these measurements vary with incidence angles the intensity, defined as the average of horizontally and vertically polarised brightness temperatures, remains almost constant in the range of 0 to 40 degrees over sea ice. By averaging over this incidence angle range we obtain

more brightness temperature data per grid point per day reducing considerably the uncertainty. The averaged product is available on a daily basis up to 85°N latitude. The data is collected for an entire day and is averaged for each grid point to provide a L3B daily mean brightness temperature product. Finally, the data is geolocated on a NSIDC polar-stereographic projection that provides grid cells with the same areal extent of 12.5 km horizontal resolution.

## 2.2 Radiative transfer models

For our analysis we selected the radiative transfer models of Kaleschke et al. (2010) and Maaß et al. (2013) to simulate brightness temperatures above sea water and ice at 1.4 GHz. The two models have been chosen because the model of Kaleschke et al. (2010) is a rather simple single-layer model that has been successfully applied for operational sea ice thickness retrieval (Kaleschke et al., 2012), whereas the model of Maaß et al. (2013) consists of multiple layers. Both models provide brightness temperatures as a function of the considered layers' temperature, thickness and permittivity.

The radiative transfer model of Kaleschke et al. (2010) describes the microwave emission of a dielectric slab of a single layer of sea ice with a semi-infinite layer of air on top and a semi-infinite layer of ocean water below (the model is further referred to as KA2010). To obtain a solution converging to the brightness temperature of open water for close to zero ice thickness, Kaleschke et al. (2010) introduces a parameterization for the sea ice thickness variability used for semi-incoherent averaging. As long as the root mean square thickness variations of the illuminated footprint are much larger than the electromagnetic wavelength used in the model, it is possible to average the emissivity over a range of sea ice thicknesses assuming that all coherent propagation effects are averaged out. The model does not include a dynamic snow layer; however, the presence of snow influences the simulated brightness temperatures through the effect of thermal insulation of the upper sea ice layer.

The radiative transfer model from Maaß et al. (2013) is based on radiative transfer equations and describes the upwelling brightness temperature at h and v polarisation from snow and sea ice represented by plane-parallel layers without surface roughness (referred as MA2013). The model extends the radiative transfer model of Burke et al. (1979) with assumptions of snow and sea ice permittivities. In our simulations the MA2013 model consists of three layers of sea ice and one layer of snow on top of the ice. All ice layers have the same properties except for the ice temperature that linearly changes between the lowest layer bordering the ocean and the upper layer facing the atmosphere. The snow is assumed to be dry and has a snow density of $\rho_{snow} = 330 \, [kg/m^3]$ that accounts for the climatological average value for Arctic average snow density over sea ice in March (Warren G. et al., 1999). In contrast to the model of Kaleschke et al. (2010), the snow layer does not only affect the temperature of the underlying sea ice, but also the radiation incidence angle. As an addition to the model described in Maaß et al. (2013) we take into account multiple reflections within sea ice instead of only considering first order reflectivity.

Both models consider the sea ice thickness subpixel-scale heterogeneity of open ocean and sea ice with a statistical ice thickness distribution obtained by observations (As used in Algorithm II* by Tian-Kunze et al. (2014)). We calculate the brightness temperatures for ten linearly divided sea ice thickness bins with a maximum of 1 meter thickness in order to represent typical first year sea ice. Then, the brightness temperature is the average of the ten respective bins weighted by the sea ice thickness distribution.

Sea water emissivity calculations are based on Fresnel equations with the descriptions of sea water after Ulaby et al. (1981) with permittivities obtained by Klein and Swift (1977). Wind induced sea surface roughness influences are assumed to be small and will be neglected (Dinnat et al., 2003). To correct for galactic background radiation and atmospheric deviations an atmospheric model (Peng et al., 2013) is taken, forced by climatological mean from 65 years of NCEP data (Kalnay and Kanamitsu, 1996). The cosmic contribution to the overall brightness temperatures is set to 2.7 K. In these simulations, we restrict the brightness temperature calculation to nadir incidence angle. The freezing temperature of sea water is set to -1.8°C.

## 2.3 The ORAP5 reanalysis

The radiative transfer models are forced using data from the Ocean ReAnalysis Pilot 5 (ORAP5) project, which is provided by the European Centre of Medium range Weather Forecast (ECMWF) (Tietsche et al., 2014). The sea ice thickness and snow depth, the surface and sea water temperature, the sea ice fractional coverage and the sea surface salinity are taken from ORAP5 data. The reanalysis has been produced using the NEMO global ocean model version 3.4, which was run on the DRAKKAR ORCA025.L75 configuration for 34 years, covering the years from 1979 to 2013. The configuration uses a tripolar mesh grid with poles located in Greenland and Central Asia in the northern hemisphere, as well as a pole in the Antarctic in the southern hemisphere. The spatial resolution ranges from 1/4 degree at the equator to a couple of kilometers in the polar regions with 75 vertical levels in the ocean. The atmospheric forcing fields are derived from the ERA-Interim reanalysis (Dee et al., 2011).

The dynamic-thermodynamic Louvain-la-Neuve Sea Ice Model second generation (LIM2) has been coupled to the NEMO ocean model (Bouillon et al., 2009). Sea ice is represented with a two-dimensional viscous-plastic rheology that interacts with the atmosphere and the ocean. A simple three-layer model (one for snow and two layers for ice) is used to determine sensible heat storage and vertical heat conduction. Vertical heat fluxes are calculated based on the thermodynamic energy balance according to Semtner (1976). The sea ice thickness is determined by the surface balance of radiative, turbulent and heat fluxes and the conductive heat balance between the bottom part of the sea ice and the ocean. Snow is accumulated by solid precipitation in case sea ice is present. If the surface temperature of the snow-ice system exceeds freezing temperature the surface temperature keeps unchanged at the freezing point and the remaining energy is put into melting of snow and afterwards sea ice. The albedo is a function of the snow and ice thickness, the state of the surface and the cloudiness. Sea ice coverage is derived by the surface energy balance over open water, the contribution of closing leads and the Operational SST and Sea Ice Analysis (OSTIA) system, which assimilates sea ice concentration from the Satellite Application Facility on Ocean and Sea Ice (OSI-SAF) dataset produced by the European Organization for the Exploitation of Meteorological Satellites (EUMETSAT).

This study focuses on the winter season in 2012/2013, more precisely on November 2012 and March 2013. The period has been chosen due the availability of the reanalysis dataset ORAP5 and SMOS measurements (v. 5.05). As a pilot-project with the goal to deliver ocean reanalysis with highest quality standards the dataset is not yet operational and therefore further processed. The link between brightness temperatures and sea ice thicknesses is established in low temperatures (Kaleschke et al., 2010) due to the ice permittivity's dependency on brine volume fraction, which in turn depends on ice temperature and salinity (Pounder, 1965; Cox and Weeks, 1973). The connection between the ice permittivity on the brine volume fraction is the basis of the sea ice thickness retrieval and will be of the sea ice thickness assimilation. November and March are the first and

**Table 1.** Uncertainties of the ORAS5 reanalysis and monthly variations of the ORAP5 reanalysis for the radiative transfer model input parameters expressed as the 99% quantile.

| No. | Model parameter | ORAS5 uncertainty | | ORAP5 monthly variation | |
|-----|-----------------|----------|----------|----------|----------|
| | | Nov 2012 | Mar 2013 | Nov 2012 | Mar 2013 |
| 1 | Sea ice thickness [m] | 0.24 | 0.17 | 0.76 | 0.87 |
| 2 | Sea ice concentration [%] | 4.4 | 8.1 | 97 | 69 |
| 3 | Sea ice temperature [K] | 0.31 | 0.87 | 18.5 | 18 |
| 4 | Snow depth [m] | 0.03 | 0.03 | 0.1 | 0.17 |
| 5 | Sea surface salinity [$g * kg^{-1}$] | 0.38 | 0.32 | 3.56 | 1.5 |

the last month in which temperatures are below freezing in the winter season (Vikhamar-Schuler et al., 2016) and are therefore chosen.

As the ORAP5 reanalysis does not provide uncertainties on its own, we use the uncertainties from the follow-on product ORAS5 reanalysis (Zuo et al., in preparation). The uncertainty values listed in table 1 represent the deviation of 99% of all values in an area north of 50°N over first year ice (1 m and below). We use the 99% quantile to exclude outliers and find a representative value for the majority of grid cells. The same statistical quantity is used for the seasonal variation of changing physical property between the beginning and the end of the month for November and March.

## 2.4 Brightness temperatures bias correction above sea water

To investigate the quality of the radiative transfer models for sea ice areas in the ORAP5 setup we want to keep the brightness temperature difference above sea water as small as possible. Thus, we first check the representation of brightness temperatures over open ocean of the models. Both models use the same equations to calculate the emissivity of water areas based on Klein and Swift (1977) and will thus produce the same brightness temperatures using same input data. Therefore we here only show the correction for MA2013. L-Band brightness temperature variations in Arctic open waters are low compared to sea ice and the difference between SMOS measurements and simulated brightness temperatures from the radiative transfer models should be less than 2 K assuming temperatures around freezing point and 30 psu salinity (Berger et al., 2002).

We simulate brightness temperatures in all open water areas north of 50° latitude. As a first step, we project the ORAP5 reanalysis on the polar-stereographic grid SMOS is using. Afterwards, we obtain a monthly average by calculating brightness temperatures for each day of the month using daily input data. Then, we average all brightness temperatures corresponding to a single day to a monthly value. We find an average bias of 4.5 K between MA2013 and the SMOS observations in November and March (Fig. 1). To identify the open water areas, we exclude all data points with a fractional sea ice coverage above zero in the ORAP5 reanalysis and also exclude all data points flagged as land, either in the reanalysis product or SMOS observations. Furthermore, brightness temperatures of more than 120 K are considered as outliers and are excluded as well. Finally, a total of 99085 data points show an average open water brightness temperature $\overline{TB}_{SMOS} = 100.7K$, whereas the models have an average of $\overline{TB}_{model} = 96.1K$.

To correct for the bias of open water areas we add the difference of 4.5 K to the overall brightness temperature of sea water. Subsequently, results of the radiative transfer models show the main accumulation of data points at around 99 K and a second, weaker one beginning at 99.5 K, each one with a tail towards higher brightness temperatures of SMOS. The wide range of observed brightness temperatures at 99 Kelvin is explained complexity of nature and the simplicity of the radiative transfer

5   model. As not all parameters are taken into account in the model, e.g. wind speed, the simulated brightness temperature tend to be less diverse. The second tail at 99.5 K has evidence in the Baltic Sea. In that area, lower sea surface salinity and higher water temperature compared to the rest of the Arctic waters leads to higher brightness temperatures. However, this is not a general statement about the quality of the brightness temperatures above Arctic sea water but a correction only for this analysis. The correction is used in all following simulations using the radiative transfer models.

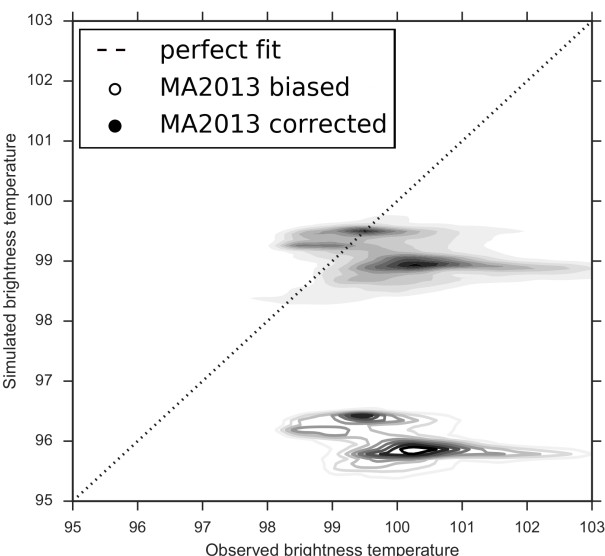

**Figure 1.** Sea water brightness temperature comparison between SMOS and MA2013. Contour lines (MA2013 biased) represent a direct comparison between simulated and measured brightness temperature above open sea water. Filled contours (MA2013 corrected) represent the same comparison but with a correction for simulated brightness temperatures of 4.5 K.

10   **2.5   Sea ice growth model**

To obtain a reference point for Arctic sea ice thickness growth for the radiative transfer model sensitivity study in 3.2, we utilize an empirical sea ice growth model (Lebedev, 1938). The sea ice thickness increase is parameterized by $d = 1.33\Theta^{0.58}$ [cm] with the freezing days $\Theta = \int (T_f - T_a)dt$ as a function of the freezing point of sea water $T_f \approx -1.9°C$ (Maykut, 1986) and the surface air temperature $T_a$ [in °C]. The sea ice growth model has been used in various previous studies (e.g. Yu and

15   Lindsay, 2003) and will provide an initial estimate of the sea ice thickness growth over a certain time period.

## 3 Results

### 3.1 Brightness temperature comparison

Brightness temperatures simulated with MA2013 are generally higher than brightness temperatures of KA2010 by up to around 15 Kelvins (Fig. 2). The largest differences are located in the outer sea ice zones with the highest magnitude where sea ice concentration is close to 100%, and an increase of sea ice thickness with time is expected, such as in the East Siberian Sea or the Canadian Arctic Archipelago in November, or the Sea of Okhotsk in March. For a first evaluation of the brightness temperature models, the two extreme cases of open water and 100% multi-year thick sea ice can be considered. Since we already treated the lower boundary of open ocean brightness temperatures with a water bias correction as indicated above, we now concentrate on the upper boundary of a saturated signal over thick sea ice areas. We find higher simulated brightness temperatures in the central part of the Arctic by MA2013. The value in MA2013 saturates around $\sim$255 K whereas KA2010 shows a maximum $\sim$240 K. There are no indications for seasonal changes between brightness temperatures from November and March except the increased area covered by sea ice. The variability in March shows a larger variation of brightness temperature difference close to the sea ice edge.

Brightness temperatures measured by SMOS appear to be in between the simulated ones of KA2010 and MA2013 influenced by strong spatial differences (Fig. 3). In November 2012, both models show higher brightness temperatures in the East Siberian Sea and the Canadian Arctic Archipelago. Lower brightness temperatures are located in the Canadian Basin and the Chukchi Sea with extension to the Bering Street. A different picture is shown in the central Arctic at grid points with more than 80% sea ice concentration coverage, where both models simulate brightness temperatures with deviations of opposite directions. In this area, the model of KA2010 shows lower ($-2.37 \pm 7.28$K) and MA2013 higher ($9.38 \pm 7.46$K) brightness temperatures compared to SMOS observations in November 2012. The same is true for March 2013, although KA2010 ($0.30\pm7.13$K) shows a stronger agreement in the central Arctic than MA2013 ($10.84\pm7.06$K). Brightness temperature deviations between KA2010 and the SMOS measurements show a higher variability between positive and negative differences. MA2013 appears to be positively biased not only in November 2012, but also in March 2013. The simulations exceed SMOS brightness temperatures almost everywhere in the Arctic with deviations up to 20 K in the Labrador Sea and Sea of Okhotsk.

A comparison between all simulated and modeled brightness temperatures in the Arctic shows that 92% data points are located at 105$\pm$3 K and 240$\pm$7 K (Fig. 4). The latter is associated with thicker sea ice related to saturated brightness temperatures at 1.4 GHz. Simulated brightness temperature shows a high correlation of the distribution state of $r = 0.98$ or $r = 0.97$ with SMOS measurements. KA2010 brightness temperatures are on average 2 K lower than SMOS in November, whereas MA2013 simulates larger brightness temperatures in thick ice regions in March and November. Furthermore, as already seen in figure 3, MA2013 overestimates brightness temperatures above $\sim$190 Kelvins also in March. The difference between simulated and observed brightness temperatures is largest in between the main clusters, although most points appear to concentrate around the 1:1 line.

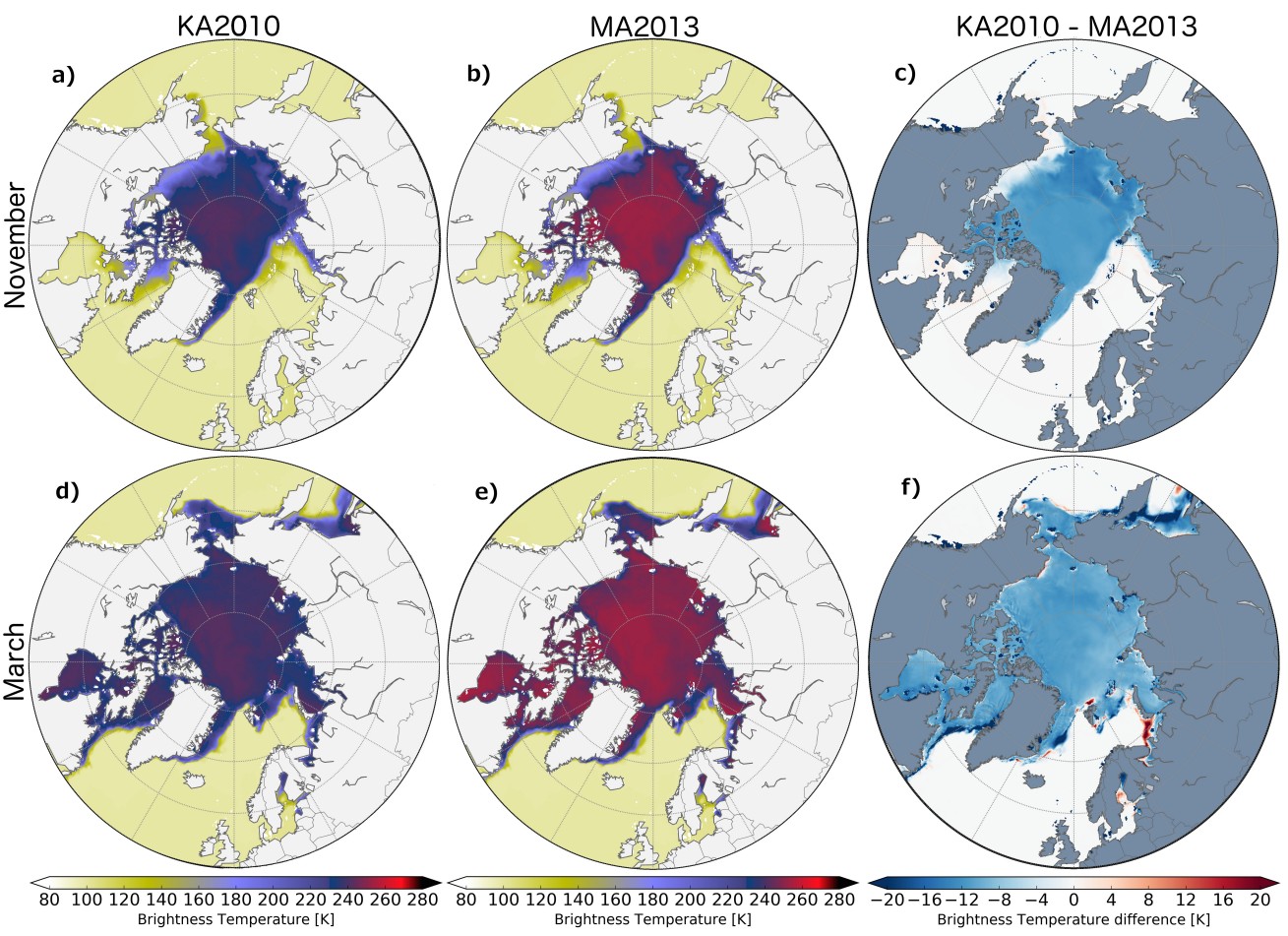

**Figure 2.** Monthly brightness temperatures simulated by KA2010 (left column) and MA2013 (center column). (**a-c**) shows the November 2012 brightness temperature distribution based on ORAP5 reanalysis input data with a comparison plot of both models at (**c**). (**d-f**) is equal, but for March 2013.

### 3.2 Radiative transfer model sensitivity study

In order to identify the most important input variables for the radiative transfer models, we evaluate the sensitivity of the models to certain changes of sea ice, snow and sea water parameters. We keep all but one parameter fixed at a monthly value and calculate the brightness temperatures for the minimum and the maximum simulated value within the month for one physical

5   parameter. That will give us two different brightness temperatures, one for the minimum, one for the maximum, of which the difference is the range of brightness temperature change related to one of the parameters that can be expected. Varying all input parameters provided by the ORAP5 reanalysis we quantify the impact of certain physical parameters on our brightness temperatures at a specific place over the time span of one month.

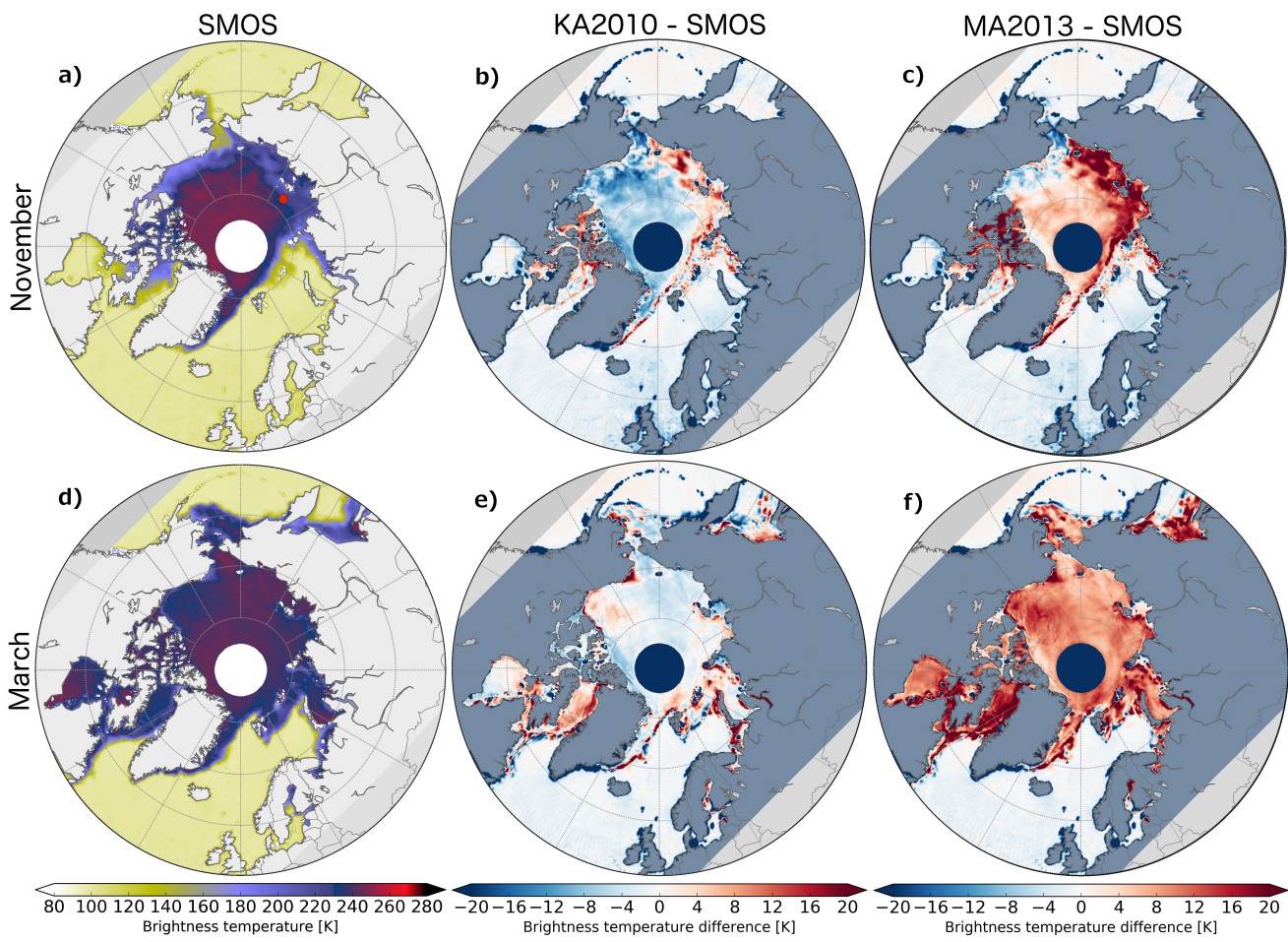

**Figure 3.** SMOS brightness temperatures (**a,d**) compared with simulated brightness temperatures using KA2010 (**b,e**) and MA2013 (**c,f**) in November 2012 (**a-c**) and March 2013 (**d-f**). The red dot in (**a**) indicates the position of the investigated grid cell in the Laptev Sea.

The most important input parameters for brightness temperature calculations with the radiative transfer models are the sea ice fractional coverage, sea ice thickness and sea ice temperature (Fig. 5, accounting for 92% grid points in March and November). For both seasons, the sea ice temperature has the largest impact on the brightness temperatures in the central Arctic with the largest spatial extent in MA2013. In March when the Arctic sea ice extent reaches its maximum, the ice temperature is the most important parameter in about 25% of the area of the Arctic. Closer to the outer sea ice regions, the sea ice fractional coverage is most influential for the largest part of the Arctic sea ice (around 60% of the total sea ice area). During the sea ice growth season in November, the leading impact of sea ice fraction extends all the way to the coastal areas in the East Siberian Sea, whereas the Canadian Basin is dominated by sea ice thickness growth. In any case, the sea ice thickness is most important close to the sea ice edge (25% of the area in November). The impact is lower when the sea ice thickness is predominantly thicker than half

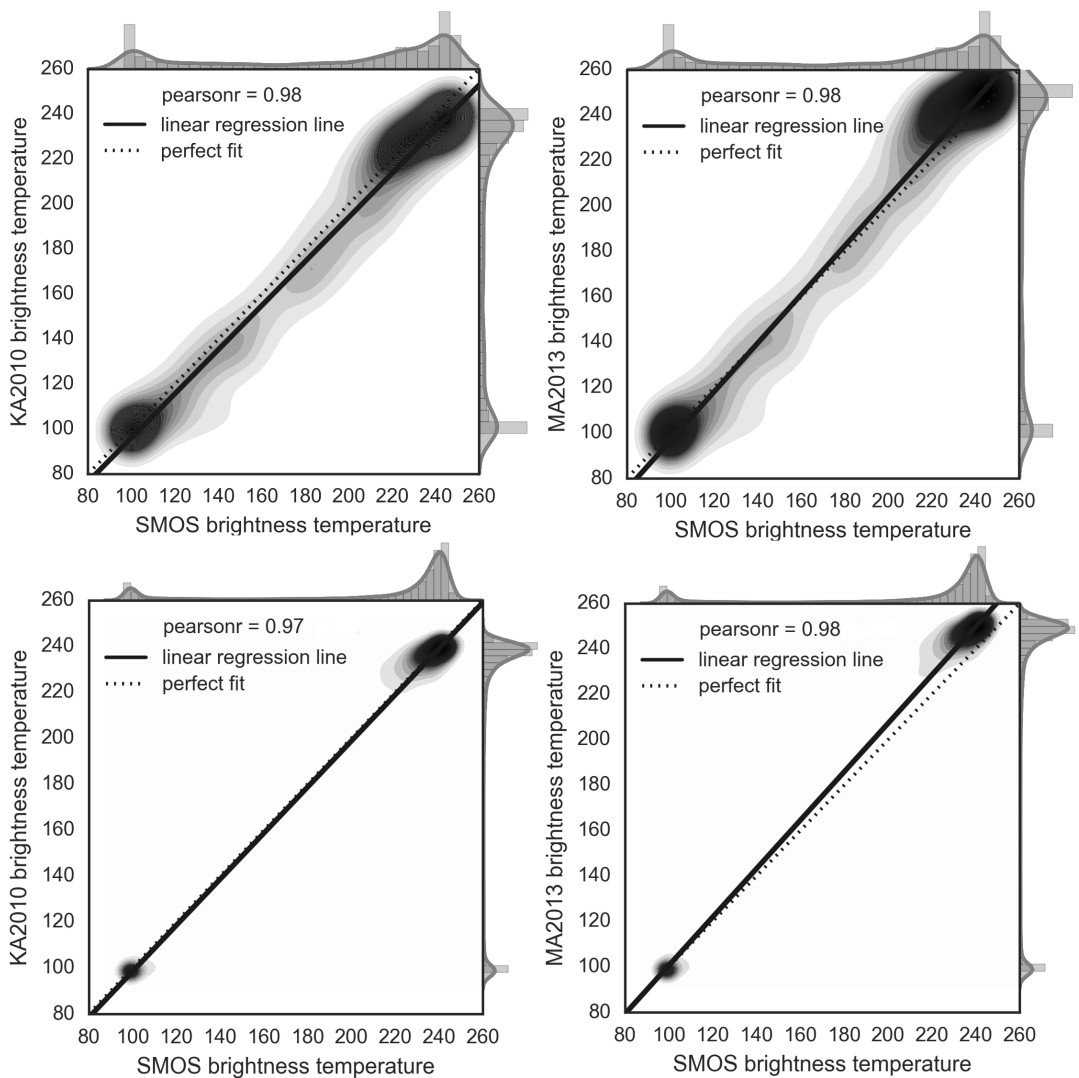

**Figure 4.** Brightness temperature comparison between simulated and observed brightness temperatures in November 2012 (top) and March 2013 (bottom) for KA2010 (left) and MA2013 (right). The pearson correlation r between simulated and observed brightness temperatures over sea ice for MA2013 and KA2010, respectively is stated in the legend.

a meter and exceeds SMOS sensitivity (5% in March). However, the effect of sea ice concentration and thickness is similar in both models. In the very outer marginal sea ice zone it appears that sea surface temperature dominates (7%). The sea surface salinity only contributes in very small areas in the Fram-Strait where the sea surface temperature and salinity are higher (<1%).

The propagating errors from ORAP5 uncertainties to the brightness temperature simulations are shown in table 2. Similar to the method used in fig. 5 one parameter varies over a range of values whereas all other parameters are fixed to a default value. The range of values is set to be the ORAS5 monthly uncertainty of November (Table 1). At that time, the sea ice thickness

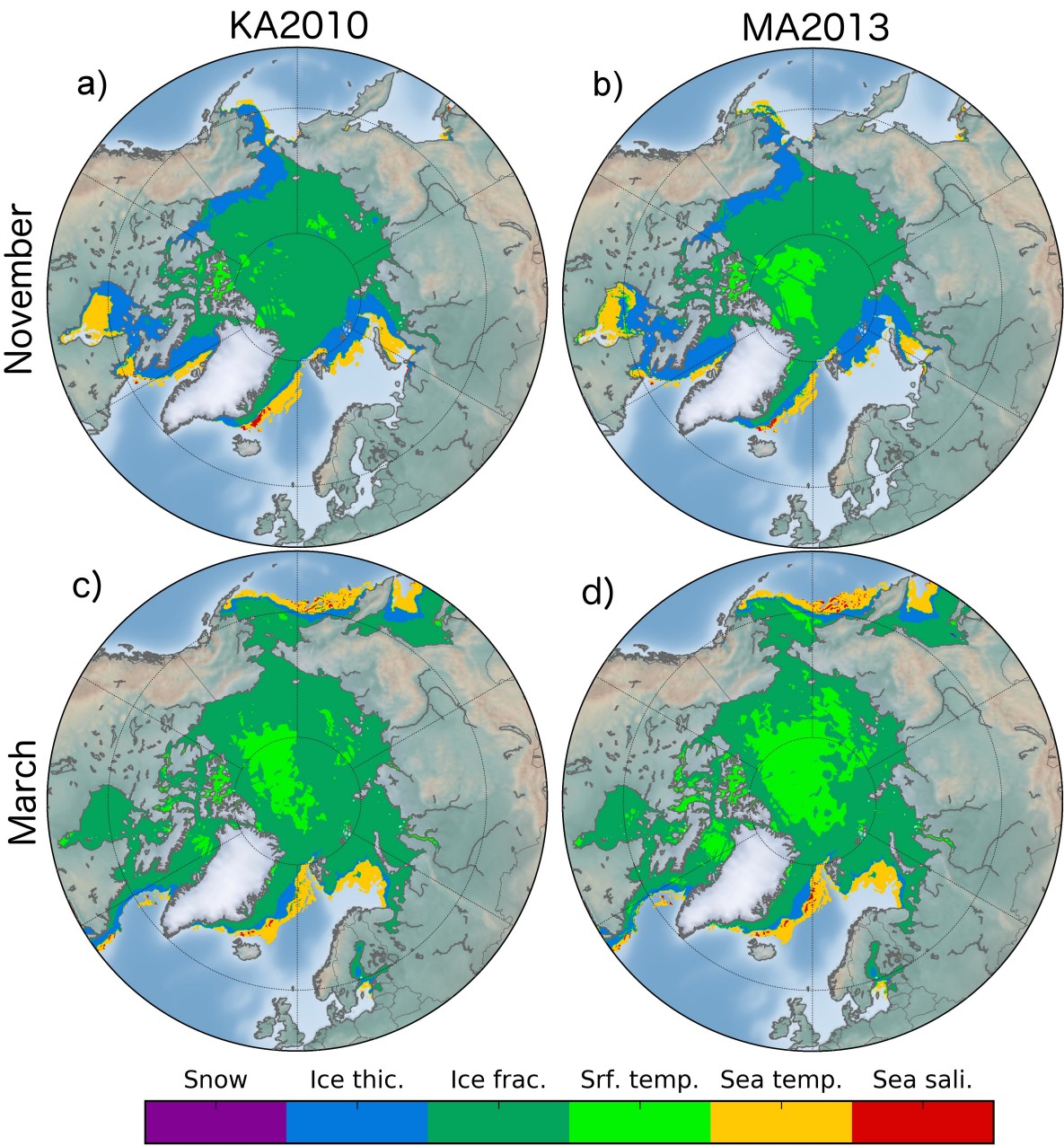

**Figure 5.** Most influencial physical variables from ORAP5 on the brightness temperature gradient. Monthly values are shown for November 2012 (**a-b**) and March 2013 (**c-d**) for KA2010 (**a-c**) and MA2012 (**b-d**).

uncertainty is higher than in March as it is in the middle of the sea ice growth season. This calculation is performed twice, once for thin sea ice (10 cm) and once for a thick sea ice (50 cm) (except in case of varying sea ice thickness). For both sea

**Table 2.** Changes of simulated brightness temperatures in KA2010 and MA2013 related to alternating physical parameters. The range is based on the ORAS5 uncertainties of November (Tab. 1). Two cases of thin sea ice (10 cm) and thicker sea ice (50 cm) are shown.

| No. | Parameter | ΔTB at 10cm in [K] | | ΔTB at 50 cm in [K] | | Default value | Range |
|---|---|---|---|---|---|---|---|
| | | KA2010 | MA2013 | KA2010 | MA2013 | | |
| 1 | Sea ice thickness | 120 | 152 | 120 | 152 | 0.1 [m]/ 0.5 [m] | 0 - 0.24 [m] |
| 2 | Sea ice concentration | 4 | 6 | 8 | 8 | 100 [%] | 95.6 - 100 [%] |
| 3 | Sea ice temperature | 0 | 0 | 0 | 1 | -5 [$^\circ$C] | -6 - -5 [$^\circ$C] |
| 4 | Snow depth | 3 | 4 | 1 | 1 | Ice thickness * 0.1 | 0 - 0.03 [m] |
| 5 | Sea surface salinity | 0 | 0 | 0 | 0 | 30 [$g*kg^{-1}$] | 29.8 - 30.2 [$g*kg^{-1}$] |
| 6 | Sea ice salinity | 2 | 7 | 15 | 12 | 8 [$g*kg^{-1}$] | 4 - 12 [$g*kg^{-1}$] |

ice thickness values, the ORAP5 sea ice thickness uncertainty of 24 cm dominates the simulated brightness temperature signal with a value up to 152 K in MA2013. The influence of all other physical properties do not exceed more than 8 K except the sea ice salinity with values up to 15 Kelvin. The effect of the sea ice temperature and sea surface salinity vanishes in this case of thin sea ice.

For a more detailed analysis on the contribution of sea ice thickness and concentration to the modelled brightness temperatures, a sample freeze-up situation is investigated at a point located in the Laptev Sea (77.5°N, 137.5°E) from October to November 2012 (Fig. 6). The observational sea ice concentration product ASI (ARTIST Sea Ice algorithm) (Kaleschke et al., 2001; Spreen et al., 2008) shows a rapid freeze-up to $80\%$ sea ice coverage in just a few days. The brightness temperatures of SMOS measurements and the KA2010 and MA2013 models show high agreement with some exceptions on the first days

of the freezing period that starts around the 25. October. The simulated brightness temperatures appear to underestimate the SMOS measurements at the beginning of the season which leads to a more linear brightness temperature increase rather than a logarithmic shape as observed from the SMOS measurements. However, the simulated sea ice concentration of ORAP5 is lower than the observed ASI sea ice concentration and needs almost two weeks to catch up to the same coverage as ASI. The sea ice thickness on the other hand shows a fast thickening to more than half a meter even before the main freeze-up event takes

place. The sea ice growth model of Lebedev (1938) accumulates sea ice as a function of the temperature difference between the surface air temperature and freezing point of water, as well as the number of freezing days below zero degrees. In contrast to the sea ice thickness of ORAP5, Lebedevs' parameterization shows a gradual increase of ice thickness throughout the freeze-up event. Following, we observe an underestimation of sea ice concentration and an overestimation of sea ice thickness in the reanalysis over a range of two weeks, whereas the SMOS brightness temperature only deviate more than 20 K for 5 days.

**4   Discussion**

Our results indicate that brightness temperature differences up to around 15 K, and even higher differences at the ice edge, can be due to the usage of different radiative transfer models (Fig. 2). Even though both models tend to have the same signatures,

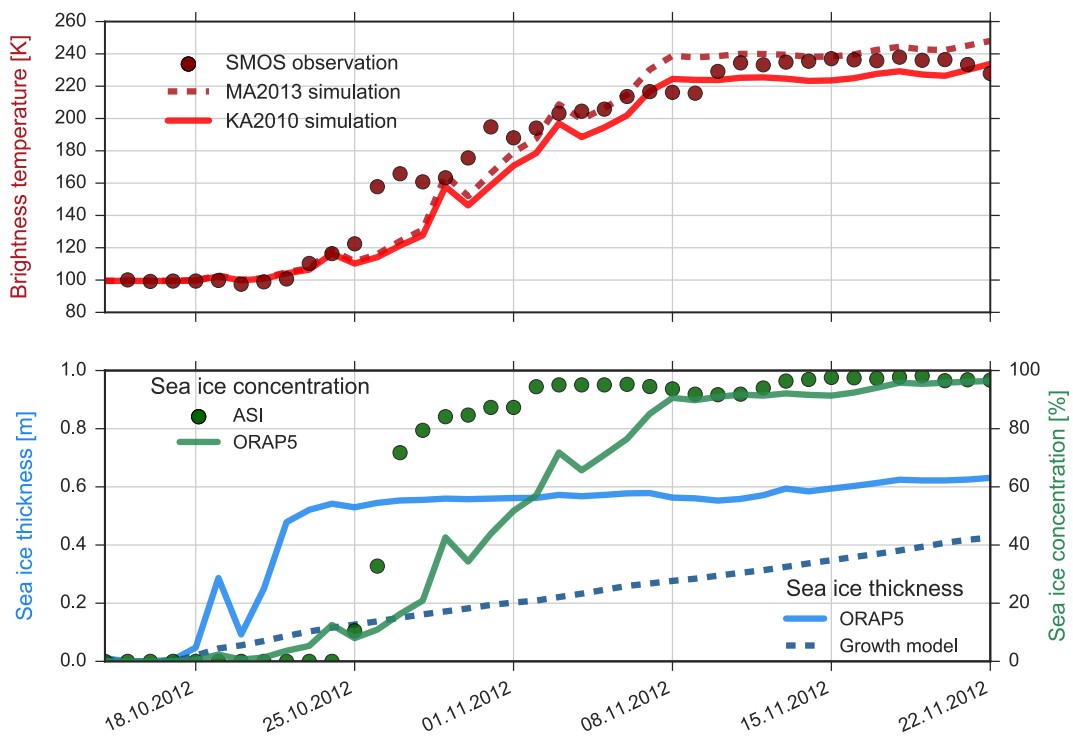

**Figure 6.** Freeze-up event in October to November 2012 that took place in the Laptev Sea (77.5 N, 137.5 E). Brightness temperature time-series of from KA2010, MA2013 and SMOS measurements (top) and sea ice thickness and concentration time-series of ORAP5, ASI and Lebedev retrieval model data (bottom).

KA2010 shows lower brightness temperatures than the MA2013 in the whole Arctic. This was expected as the MA2013 model is able to take multiple sea ice layers into account, as well as the radiometric effect of snow on top of sea ice, whereas KA2010 only indirectly includes the effect of snow with the representation of the thermodynamic insulation effect. However, compared with SMOS brightness temperatures, it appears that MA2013 driven with the ORAP5 output overestimates brightness

5   temperatures in many parts of the Arctic, most pronouncedly in March in the central Arctic region. This area is mostly covered by thick sea ice. We cannot assimilate SMOS data here because microwave radiation does not contain information on the thickness of thicker sea ice. In contrast, KA2010 shows good agreement in the central Arctic area. For brightness temperature assimilation purposes that would be clearly beneficial as the main assimilation should take place in regions with thin sea ice rather than in the central Arctic. Therefore, a more comprehensive representation of sea ice with radiative transfer equations

10  using multiple layers does not necessarily need to be an advantage for brightness temperature assimilation.

    The analysis shows spatial differences between SMOS and simulated brightness temperatures throughout the Arctic with largest differences in the outer Arctic regions for November 2012 and March 2013. However, the sign of the deviation changes according to the region. In the East Siberian Sea both models simulate higher brightness temperatures compared with SMOS,

whereas lower values are shown in the Canadian Basin. In the latter the fractional sea ice coverage increased from almost open water up to an average value of 60% in November, whereas the coverage in the East Siberian Sea stayed more or less constant at 100% (not shown here). To estimate the sea ice thickness from brightness temperature measurements, the sea ice concentration becomes more important the thicker the sea ice gets.The reasons are: 1) The brightness temperature difference

between ice and water is higher for thicker ice, thus a changing ice concentration changes the mixture's brightness temperature more than for thinner ice. 2) Due to the lower sensitivity of the L-band signal to ice thickness for thicker ice, the brightness temperature difference induced by changing ice concentration leads to a higher ice thickness difference for thicker ice than for thinner ice. (e.g. Kaleschke et al. (2012)). That makes it necessary to use a high-precision auxiliary sea ice concentration product to account for brightness temperature changes due to alteration in sea ice concentration.

Here, we provide sea ice fractional coverage from our reanalysis to the radiative transfer models and are thus able to reduce the uncertainty of sea ice fraction changes compared to studies with an assumption of a constant 100% sea ice concentration coverage (Table 2)(Tian-Kunze et al. (2014)). However, the quality of the ice concentration data is essential for the quality of L-band brightness temperature assimilations. At a sea ice thickness of around 50cm, an uncertainty of 5% fractional sea ice coverage accounts for a difference of 8 K (Kaleschke et al., 2010), which in turn can result in a sea ice thickness uncertainty of

more than 10 cm. In occasional events, the deviation of sea ice concentration from the reanalysis data to the here investigated ASI sea ice concentration can be even higher with differences up to 40% (Fig. 6). The largest error is due to the sea ice thickness with an estimated uncertainty of 24 cm for ORAP5 in November 2012corresponding to a brightness temperature difference of 120 K and 152 K for thin ice (below 24 cm) in KA2010 or MA2013, respectively. This uncertainty is more than 10 times higher than all other uncertainties in case of typical first year sea ice represented by the default values in table 2 except for sea

ice salinity which is a function of sea ice thickness and sea surface salinity (Ryvlin, 1974). This is benefical for assimilation purposes as 93% in MA2013 or 90% in KA2010 of all brightness temperature deviations between SMOS and the radiative transfer models are rooted in the sea ice thickness (compare table 2).

In order to assimilate thin sea ice thickness it is crucial to understand the impact of all physical parameters on the brightness temperature simulations. Kaleschke et al. (2012) found sea ice concentration and thickness changes in thin sea ice areas are

the most important variables for L-Band brightness temperatures. We here support this evidence as our most dominating dependencies for brightness temperature simulations are found to be the same throughout the Arctic (Fig. 5). Over thick sea ice in the central Arctic we find the sea ice/snow surface temperature to be the most influential parameter. Since sea ice concentration is close to 100% and sea ice thickness is above the L-Band sensitivity, brightness temperature changes are due to the impact of snow depth and sea ice/snow temperature changes that come with it. In thin sea ice regions, variations of sea

ice temperature, snow depth and sea surface salinity can have an accumulated influence of up to more than 25 K (not shown).

Our results also show a significant influence of sea surface temperature and salinity in areas of thin sea ice close to the ice edge. This is explained by a fractional sea ice coverage of less than 10%, where brightness temperature variations are dominated by changing open water emissivities. We point out that sea surface temperature and salinity get more important in regions with lower sea ice coverage. Therefore, in case partially ice-covered areas are taken into account we caution that a

climatology of sea surface temperature or salinity might not be sufficient enough to picture the transition between open water

towards the sea ice edge. This is especially true for the declining sea ice observed in the recent years as the sea ice edge is likely to be located at a different location than in the previous years.

A comparison of SMOS and simulated brightness temperatures showed a Pearson correlation of 0.97- 0.98. However, most of the data points are located at brightness temperatures for either the saturated case at 240/255 K for KA2010/MA2013 or open water areas at around 100 K (Fig. 4), especially in March. The reason is that 92% of all simulated data points over sea ice are larger than 220 K. The overall performance in terms of the range of simulated brightness temperatures over sea ice is explained by the Kolmogorov-Smirnov test ($\alpha = 0.1$). The test determines the accordance of two different datasets without making any assumption about the distribution of the data (Sachs and Hedderich, 2006). In the K-S-test, $1 - \alpha$ is the probability that two data sets originate from the same distribution, or in other words, $\alpha$ is the confidence to accept a hypothesis. Here, the test only accepts the brightness temperature distribution from March 2013 of KA2010. Therefore, it is most important for the model to agree with the saturated case in order to determine reasonable areas for brightness temperature assimilation in that KA2010 in March agrees most. We specifically concentrate on the saturated case, as it is next to water the only reliable reference point we can address for a quality assessment of the models. Thus, based on ORAP5 reanalysis input data and electromagnetic formulations used here, we suggest to use the KA2010 radiative transfer model for brightness temperature assimilation. However, for the remaining 8% of all simulated data points with intermediate sea ice thickness and concentration we are unable to find a favourable radiative transfer model as the results of the models are superimposed by the sea ice thickness uncertainty. Note, that this is no statement about the quality of the radiation model in general, than rather a suggestion for the specific assumptions and characteristics of the LIM2 sea ice models that were used in this study.

Although the statistical representation of brightness temperatures is well captured, we find large discrepancies in times of rapid sea ice changes (Fig. 6). For an example case, ORAP5 appears to have difficulties to simulate freeze-up events, in which we see an overestimation of sea ice thickness and an underestimation of sea ice concentration. The assimilation of OSI-SAF sea ice concentration into ORAP5 pushes the sea ice concentration into the right direction, but appears to be too slow to picture changes in a short period of time. A smaller fractional sea ice concentration and an overestimation of sea ice thickness then lead to simulated brightness temperatures that fit with observed SMOS brightness temperatures, even though both parameters are divergent at this time. In this case, a SMOS brightness temperature assimilation in ORAP5 may not bring much benefit as we cannot distinguish between the influence of sea ice concentration and thickness. An auxiliary sea ice concentration data product is needed to correct brightness temperature calculations for possible differences in sea ice cover. We therefore suggest a combined assimilation of brightness temperatures covering a broad spectral range from 1.4 to 37 GHz yielding information on both, sea ice thickness and concentration.

All brightness temperature calculations rely on the quality of the ORAP5 data as the results are highly influenced by uncertainties of the input parameters. A comprehensive sea ice concentration and sea ice thickness comparison between ORAP5 and observational products was made by Tietsche et al. (2015) who found an agreement of sea ice concentration in ORAP5 and OSTIA in the order of magnitude of 5% root mean square deviation. Assuming this is an optimistic estimation as OSTIA sea ice concentration is assimilated into ORAP5, the uncertainty still accounts for 7-8 K, explaining up to more than 10 cm ice thickness difference in case of thicker sea ice and is therefore still critical. They point out that the representation of sea

ice thickness in thin sea ice regions needs further improvements, especially in the vicinity of the ice edge. However, the initial freeze-up sea ice thickness in the LIM2 sea ice model is set to 0.5 m leading to an overestimation of sea ice thickness in newly formed sea ice areas. Sea ice of half a meter thickness is mostly already at the maximum thickness for a brightness temperature assimilation and will lead to essential brightness temperature differences between modeled and observational data. This is a strong argument for a brightness temperature assimilation, as it might help to correct the overestimation of ORAP5 sea ice thickness in freeze-up areas.

## 5 Summary and outlook

The radiative transfer models from Kaleschke et al. (2010) (denoted as KA2010) and Maaß et al. (2013) (MA2013) are taken as a forward operator to simulate brightness temperatures at 1.4 GHz in the winter season 2012/2013 and to identify the feasibility of the models for a brightness temperature assimilation in the global ocean reanalysis product ORAP5. Using ORAP5 input data, we compared modeled brightness temperatures with SMOS observations in November 2012 and March 2013 accounting for the start and the end of the winter season, respectively.

The results of this study indicate that both models are able to simulate Arctic-wide monthly brightness temperatures. We are able to observe a similar increase of simulated and observed brightness temperatures from thin to thicker sea ice areas. Although both models show a decent fit in November, the model of Maaß et al. (2013) tends to overestimate brightness temperatures in the saturated case of thick sea ice in March with the configurations applied here. A Kolmogorov-Smirnov test thus only accepts the brightness temperature distribution of KA2010 in March by taking the SMOS observation as the reference probability distribution. All other, especially the results of MA2013 in March, are rejected. Therefore, we suggest to use the model of KA2010 for a brightness temperature assimilation into the ORAP5 reanalysis project. This suggestion is primarily based on a comparison between SMOS and simulated brightness temperatures over thick sea ice and open water and does not make a statement about the ability of the model to reproduce brightness temperatures in thin sea ice conditions.

The most important parameters for the brightness temperature calculations over thin sea ice are identified to be the sea ice thickness and sea ice coverage. This result supports the findings of other studies (e.g. Kaleschke et al. (2012)). In thicker sea ice areas the dominant parameter is the sea surface temperature since the sea ice fractional coverage is close to 100% and sea ice thickness changes do not affect the measurements at 1.4 GHz. The influence of the sea ice temperature, snow depth and sea ice salinity increases in thinner sea ice areas but will still be less than the sea ice thickness and concentration. However, the smaller the sea ice fractional coverage, the more important are the sea surface temperature and salinity. This becomes relevant at sea ice concentration below 15%, usually in small regions at the very outer sea ice edge.

The brightness temperature assimilation is expected to result in a more accurate sea ice thickness analysis than a direct assimilation of the physical parameter as the climate model provides a series of input variables to the forward operator. These variables do not need to be replaced by climatologies, parameterizations or assumptions that may inflict the results of our sea ice thickness retrieval. However, even though the sea ice thickness and concentration in ORAP5 are well constrained by observations (Tietsche et al., 2015), both show difficulties to represent a rapid freeze-up event with an underestimation of sea

ice concentration and an overestimation of sea ice thickness. That reveals the challenge to use brightness temperatures to correct for the right physical parameter and magnitude. We recommend to combine the brightness temperature assimilation for sea ice thickness with the assimilation of an independent auxiliary observational sea ice concentration product or the simultaneous assimilation of measurements taken at higher microwave frequencies, e.g. up to 37 GHz.

The assimilation of SMOS brightness temperatures appears to be a great chance for a better representation of sea ice thickness in the ORAP5 reanalysis. Substantial differences between observational and simulated brightness temperatures are found to be largest in regions of thin sea ice, in which SMOS uncertainties of the sea ice thickness retrievals are lowest (Kaleschke et al., 2010). That reveals the possibility to retrieve and correct sea ice thickness in future investigations. However, to what magnitude these results translate to other reanalysis products or climate forecasts has to be investigated.

# 6 Data availability

L3B Brightness temperatures are provided by the CliSAP-Integrated Climate Data Center (ICDC) on http://icdc.cen.uni-hamburg.de/1/daten/cryosphere/l3b-smos-tb.html (Tian-Kunze et al., 2012). The reanalysis data of ORAP5 was kindly provided by ECMWF and is freely available on http://marine.copernicus.eu/services-portfolio/access-to-products/?option=com_csw&view=details&product_id=GLOBAL_REANALYSIS_PHYS_001_017 (Zuo et al., 2015). The ORAS5 uncertainties will be soon available at http://www.ecmwf.int/en/research/climate-reanalysis/browse-reanalysis-datasets.

*Acknowledgements.* We thank Steffen Tietsche for the provision of ORAP5 reanalysis data, Andreas Wernecke who helped to implement the cosmic and galactic background radiation representation. We thank the "Science Snack" team for fruitful discussions.

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
