# Peer review of "Arctic sea ice signatures: L-Band brightness temperature sensitivity comparison using two radiation transfer models"

_The Cryosphere, 2016_

## Referee Comment (RC1) · Anonymous Referee #1 · 2 Jan 2017

**General comments:**

This study shows the potential use of brightness temperature data from the ESA SMOS mission for forecast model assimilation, to improve sea ice thicknesses in thin ice regions. Although the application of brightness temperature data rather than a derived thickness product is not a new concept, this is the first time that the usefulness of SMOS data has been explored in detail and the manuscript will be of interest to the observation and modelling communities. However, I have some concerns to be addressed.

The manuscript currently lacks a suitable level of transparency and detail regarding a.) the uncertainties associated with modelled brightness temperature and b.) the limitation of ORAP5 data in the development of a reliable brightness temperature model. The fact that the brightness temperature calculations rely on the quality of the ORAP5 data is only explicitly stated in the Discussion. This should be explained much earlier in the manuscript, and explored quantitatively.

1.) The authors state the assimilation of brightness temperature in forecast models is preferable to a sea ice thickness product due to the "assumptions, parameterizations and auxiliary data" associated with thickness retrievals. However, for comparison with observed brightness temperature data they generate two brightness temperature models that rely on multiple input sea ice parameters from the ORAP5 reanalysis (thickness, concentration, temperature etc.). These in themselves will be derived model parameters with their own inherent uncertainties. The authors should briefly explain how each of these parameters is derived in ORAP5 and state their associated uncertainties. Based on this they should expand on why using a radiative transfer model (which is itself developed from derived parameters) as a forward operator in a brightness temperature assimilation scheme for thin sea ice thicknesses is preferable to using observed thickness data.

2.) An important consideration is the relative impact that the uncertainty in each of the input ORAP5 sea ice parameters will have on brightness temperature. The authors touch on this by carrying out a sensitivity study of their radiative transfer model to identify the most important input parameters for the two models, and display the results in Figure 7. This is the truly novel part of the manuscript and needs to be expanded. An easy and effective way to do this would be to tabulate the effects of the sensitivity study for both models. What effect (expressed as a percentage, for example) does varying each parameter to its minimum and maximum simulated value have on brightness temperature over thinner (say 10 cm) and thicker (say 50 cm) sea ice? This is essentially a tabulated expansion of the explanation given on P16 L2-4.

Specific comments:
P1 L4: It is perhaps more accurate to state that SMOS brightness temperatures have been proven to be valuable in estimating modal thin sea ice thicknesses, not mean. See for example [1].

P3 L30: Why 2 m? 2 m is 0.5 m greater than the maximum SMOS validation thickness.

P5 L27: The assumption of dry snow is oversimplified. Despite this being a necessary assumption made for the model, the authors should comment on the potential impacts of a wet snow layer on brightness temperature. This is especially important, as wet snow is most common on thin FYI, such as that measured by SMOS, even in winter.

P6 L8-19: The relevance of the brief introduction to NEMO and LIM would not be clear to someone who is unfamiliar with ORAP5. I believe ORAP5 was produced from these models, but this is not explicitly stated in the manuscript.

Conclusion: A comment on the potential for a similar approach over thick ice would be useful. If brightness temperature can't be used, what could?

Technical comments:

P2 L1-2: Not great wording. Suggest change to "This capability is especially important as the Arctic Ocean shifts to a new state, in which older, thinner sea ice is being replaced by younger and thinner ice".

P7 L28, P13 L7 and P13 L7: "fairly match", "properly agree" and "decently well". What do these mean? Re-word.

P8 L16: add "the" before largest and highest

References:

[1] L. Kaleschke, X. Tian-Kunze, N. Maass, A. Beitsch, A. Wernecke, M. Miernecki, et al., "SMOS sea ice product: Operational application and validation in the Barents Sea marginal ice zone," Remote Sensing of Environment, vol. 180, pp. 264-273, Jul 2016.

TCD

---

## Referee Comment (RC2) · Anonymous Referee #2 · 14 Feb 2017

**Comments**

**General**

With this work the authors achieved insight in several aspects of modelling and observing arctic sea ice with 1.4 GHz brightness temperature data from SMOS for November 2012 and March 2013. The key to the results is the coupling of a full set of models, consisting of meteorological forcing, sea-ice physics, microwave emission and comparison with observations. In spite of some inconsistencies, especially during the sea-ice growth phase, the results are encouraging. I am glad for these results. Congratulation! Exploration on what these results mean with respect to sea-ice research would be helpful. Unfortunately the text is often unclear, sometimes misleading or erroneous, the main reason why it took me a long time for writing this response. The authors should try to find a more appropriate title and better names for each section. Furthermore they should reduce the hand-waving explanations in words, and instead use the logic of mathematical formulations. Furthermore, there is a need for improving the language.

**Special - suggestions for improvements**

1) Throughout the paper change from plural to singular (as already done in Table 2) for:
- "sea-ice thicknesses" because there is only one thickness parameter for sea ice.
- "snow thicknesses", even better, use "snow depth".
- "sea-ice concentrations", unless you distinguish between different types of sea ice (e.g. first year, multi year).

2) Section 2 "Data and Methods". This section is poor:
- formally (longest part of Section 2 without any subsection, followed by short Subsections 2.1 and 2.2, and by a related Section 3),
- logically (models are not well presented, some parameters not defined, equations are missing, Figure 1 confusing, Table 1 inconsistent with text for brine volume fraction),
- with respect to the motivation for this paper (e.g. the multilayer model description starts with "The incoherent model used in Maaß et al. (2013) is based on", without any indication why this text is found here, and the same holds for the single-layer model). Start e.g. with "For our analysis we selected ... It is useful because ..."
- and details:

p. 1, line 22: The statement "Microwave radiation is especially useful to derive thin sea ice thicknesses as it is able to penetrate snow and sea ice for more than half a meter". This statement is incorrect because it is far too general. Microwave (1 to 300 GHz) penetration into sea ice is certainly much less than half a meter in most types of sea ice and it is marginal even at the lowest frequency.

p.3, line 6: ORAP5 seems to play an important role. A description would help, a proper reference is a 'must'.

p. 3, lines 14 - 15: What do you mean with "salinity ration"? (misprint?)

p. 5, lines 3-4: Improve this part, but do not try to correct the galactic radiation nor let the atmosphere deviate. What do you mean?: "To account for corrections of the galactic background radiation and atmospheric deviations a simplified atmospheric model (Peng et al., 2013) is taken forced by..." And why do you use a simplified model? What kind of simplification? Use mathematics to show exactly what you did. Readers might want to check.

p. 5, line 17: What do you mean with "temperature insulation"? Do you mean "thermal insulation"?

p. 5, line 23 once more: "The incoherent model used in Maaß et al. (2013) is based on radiative transfer equations and describes the emitted radiation from a stratified bare soil". Please be more specific, e.g. by writing "... describes the upwelling brightness temperature at h and v polarisation from

bare soil represented by plane-parallel layers with or without surface roughness".

p. 5, line 27: The snow density selected seems to me rather high for the usually shallow snow layers found on sea ice. Why do you consider a fixed value?

p. 5, line 32 to p. 6, line 3: Modification of the model. Either describe exactly what you did or delete this part. Note that there is a risk of introducing errors.

3) Improve the description of all methods by using appropriate figures for explaining the geometry, angles, etc. as well as mathematical formulas at least for the relevant expressions to enable definitions of the coefficients mentioned (p. 3).

4) p. 7, lines 3-5: Improve physics and timing in "In the melting season, when melt ponds form on sea ice and temperatures begin to rise". Note that temperature rise is much earlier than formation of melt ponds. Explain what you mean and improve the sentence that follows: " SMOS brightness temperatures over sea ice are impossible to connect to a specific sea ice property (Kaleschke et al., 2010)". The logic to the next sentence and its meaning are not clear: "Thus, November and March are the first and the last month, respectively, with full monthly data coverage from SMOS and therefore chosen."

5) p. 7, line 15 - : Improve " chosen as values higher than that are not expected to be seen". Also improve the following sentence: " The brightness temperature product consists of vertical and horizontal polarization, which are averaged up to 40  incidence ...". I do not understand. And what do you mean with the sentence that follows? " These brightness temperatures are said to represent L-Band measurements at nadir as brightness temperature changes that are connected to the varying incidence angles are expected to cancel out each other when both polarisations are considered."

6) p. 7 "Sea water correction" What do you mean? Please do not try to correct the water! Please first explain the purpose of this section, and then improve it, especially explain what Figure 2 is supposed to show. Its legend cannot be used to understand what the data clouds mean, nor is the caption of any help. Furthermore the quality of these data is not convincing due to the poor correlation shown. And there must be a reason for the Tb correction. Try to find the error.

7) p. 8 "Brightness temperature comparison". Give a motivation to the reader for not skipping this section.  If it is a 'result' section, then please call it accordingly.

8) p. 10, line 1, also discussion p. 17-18: Explain what the "Kolmogorov-Smirnov-Test " is supposed to check and present the results properly. This is relevant because you use this information for the decision to drop one of the models used. Can you support this decision by physical arguments?

9) p. 13, Figure 5: Nice representation. However the concentration of data points near the two main spots causes problems in the interpretation. It appears that the assessment of thin ice and medium ice concentration is difficult. Think about how to improve the situation, e.g. by omitting some of the data.

10) p. 13, lines 16-17: "we observe an underestimation of sea ice concentration" underestimation by what, i.e. which model?

11) p. 15, Figure 8: Exchange the two legends in order to be close to the respective y axis, and clarify 'growth model' with respect to caption (Lebedev ?). What is the role "Lebedev" is playing here (missing in Sect 2).

---

## Author Response (AR1)

This study shows the potential use of brightness temperature data from the ESA SMOS mission for forecast model assimilation, to improve sea ice thicknesses in thin ice regions. Although the application of brightness temperature data rather than a derived thickness product is not a new concept, this is the first time that the usefulness of SMOS data has been explored in detail and the manuscript will be of interest to the observation and modelling communities. However, I have some concerns to be addressed.

We would like to thank the referee for its initial positive assessment and will now discuss each point in detail.

The manuscript currently lacks a suitable level of transparency and detail regarding a.) the uncertainties associated with modelled brightness temperature…

As there are no derived uncertainties for the ORAP5 reanalysis product we use the uncertainties of the follow-on product ORAS5 and assume that they will be of same magnitude as the uncertainties of ORAP5. Both products have the same resolution and use the sea ice model LIM2. Following this comment, we added a new table (Tab. 2) to the manuscript in section 2 [p.5]. It shows the uncertainties compared with the monthly variation of the physical properties. The uncertainties are ten times smaller than the monthly variations except sea ice thickness. Moreover, we analyze the impact of uncertainties on the brightness temperature simulation for both models in an additional table [p.13]. The results show a large impact of sea ice thickness uncertainties on the brightness temperature simulation and far less influence by all other variables.

… and b.) the limitation of ORAP5 data in the development of a reliable brightness temperature model.

This paper does not follow the purpose to identify one of the radiative transfer models to be the correct one as the uncertainties from the reanalyzes are still noticeable, especially in the intermediate range of first year sea ice thicknesses. However, we do give a recommendation for one of the models to be more suitable for brightness temperature assimilation. This recommendation however is only based on the open water case and the saturated case over very thick sea ice, where uncertainties of sea ice concentration and sea ice thicknesses do not count. As this has been misunderstood we added a clarification in the revised version and specifically note that this is the case.

… The authors should briefly explain how each of these parameters is derived in ORAP5 and state their associated uncertainties. ..

Added to the methods. [p. 4, line 25 ff.]

Based on this they should expand on why using a radiative transfer model (which is itself developed from derived parameters) as a forward operator in a brightness temperature assimilation scheme for thin sea ice thicknesses is preferable to using observed thickness data.

Please note that also "observed sea ice thickness" products from SMOS over thin sea ice rely on radiative transfer models to derive sea ice thicknesses from brightness temperatures. Therefore in any case, there is always the uncertainty of an imperfect radiative transfer model. However, the advantage of a direct brightness temperature simulation is the availability of a coherent set of input data from the reanalyzes/forecast model. This allows us, for example, to take into account the sea ice concentration in our calculations which as not been possible before (Tian-Kunze, 2014). Furthermore, snow thicknesses are directly derived in the forecast model and are not taken from a general annual climatology. Another advantage of a brightness temperature assimilation is a better traceability of the errors as all variables belong to the same dataset. However, since this point led to confusion we added another sentence to section 1. [p. 2, lines 31-33]

An easy and effective way to do this would be to tabulate the effects of the sensitivity study for both models. What effect (expressed as a percentage, for example) does varying each parameter to its minimum and maximum simulated value have on brightness temperature over thinner (say 10 cm) and thicker (say 50 cm) sea ice?

We followed this idea and added another table to the results section [p. 13, table 2]. It shows the propagating error in brightness temperatures based on the ORAS5 uncertainties. The error is expressed in Kelvin for each quantity provided by ORAP5. The dominating uncertainty (more than 80% of all variables) over first year sea ice is based on the sea ice thickness. We conclude that as beneficial for the brightness temperature assimilation for sea ice thicknesses. Most brightness temperature differences will be due to sea ice thickness and can thus be corrected, whereas the other parameters have a minor impact. Note that the sea ice concentration over first year sea ice only shows an average uncertainty of 5%.

P1 L4: It is perhaps more accurate to state that SMOS brightness temperatures have been proven to be valuable in estimating modal thin sea ice thicknesses, not mean. See for example [1].

Changed in revised version.

P3 L30: Why 2 m? 2 m is 0.5 m greater than the maximum SMOS validation thickness

We reprocessed all results with a maximum sea ice thickness of 1 meter to account for thinner first year sea ice. The changes are almost negligible that will not alter the results. However, the new figures are included in the revised version.

P5 L27: The assumption of dry snow is oversimplified. Despite this being a necessary assumption made for the model, the authors should comment on the potential impacts of a wet snow layer on brightness temperature. This is especially important, as wet snow is most common on thin FYI, such as that measured by SMOS, even in winter.

Carsey 1992 examined the influence of wet snow above saline ice on brightness temperatures (Carsey 1992, Fig. 4-26). He does not see a significant influence on brightness temperatures at 10 GHz, even more decreasing at lower frequencies. The snow moisture is described to be below 2% if the temperatures are at 268K or lower, up to 3% if the temperature is greater than 273K (Carsey 1992, Fig. 16-2). Therefore the penetration depth will be at least 1 meter and should be negligible at 1.4 GHz.

P6 L8-19: The relevance of the brief introduction to NEMO and LIM would not be clear to someone who is unfamiliar with ORAP5. I believe ORAP5 was produced from these models, but this is not explicitly stated in the manuscript.

Considered in revised version. [p. 4, lines 24]

Conclusion: A comment on the potential for a similar approach over thick ice would be useful. If brightness temperature can't be used, what could?

A common method to derive thick sea ice thicknesses is by using altimetry of e.g. ICESat or CryoSat-2 (Kwok et al. 2009, Laxon et al. 2013). By measuring the elevation of the sea ice surface above the water line (called freeboard) it is possible to estimate ice thicknesses above 1 m. However, systematic errors are introduced by using e.g. a snow thickness climatology or fixed snow density (Kwok 2014, Ricker et al. 2015). Thus, a similar approach to this study might improve the accuracy of the freeboard calculation by using the reanalyzes data as input. In future, even a synergy of SMOS thin and altimetry thicker sea ice thickness derivation might be feasible, as it already exists for the combined SMOS and CryoSat-2 sea ice thickness retrievals (Ricker et al. 2017).

Carsey, F. D. (1992). *Microwave remote sensing of sea ice*. American Geophysical Union.

Kwok, R.: Simulated effects of a snow layer on retrieval of CryoSat-2 sea ice freeboard, Geophysical Research Letters, 41, 5014–5020, doi:10.1002/2014GL060993, http://dx.doi.org/10.1002/2014GL060993, 2014.

Kwok, R., Cunningham, G. F., Wensnahan, M., Rigor, I., Zwally, H. J., and Yi, D.: Thinning and volume loss of the Arctic Ocean sea ice cover: 2003-2008, J. Geophys. Res., 114, doi:10.1029/2009JC005312, 2009.

Laxon, S. W., Giles, K. A., Ridout, A. L., Wingham, D. J., Willatt, R., Cullen, R., Kwok, R., Schweiger, A., Zhang, J., Haas, C., Hendricks, S., Krishfield, R., Kurtz, N., Farrell, S., and Davidson, M.: CryoSat-2 estimates of Arctic sea ice thickness and volume, Geophysical Research Letters, 40, 732–737, doi:10.1002/grl.50193, http://dx.doi.org/10.1002/grl.50193, 2013.

Ricker, R., Hendricks, S., Perovich, D. K., Helm, V., and Gerdes, R.: Impact of snow accumulation on CryoSat-2 range retrievals over Arctic sea ice: An observational approach with buoy data, Geophysical Research Letters, 42, 4447–4455, doi:10.1002/2015GL064081, http://dx.doi.org/10.1002/2015GL064081, 2015GL064081, 2015.

Ricker, R., Hendricks, S., Kaleschke, L., Tian-Kunze, X., King, J., and Haas, C.: A Weekly Arctic Sea-Ice Thickness Data Record from merged CryoSat-2 and SMOS Satellite Data, The Cryosphere Discuss., doi:10.5194/tc-2017-4, in review, 2017.

**Tc-2016-273 – Second review**

**General**

With this work the authors achieved insight in several aspects of modelling and observing arctic sea ice with 1.4 GHz brightness temperature data from SMOS for November 2012 and March 2013. The key to the results is the coupling of a full set of models, consisting of meteorological forcing, sea-ice physics, microwave emission and comparison with observations. In spite of some inconsistencies, especially during the sea-ice growth phase, the results are encouraging. I am glad for these results. Congratulation! Exploration on what these results mean with respect to sea-ice research would be helpful. Unfortunately the text is often unclear, sometimes misleading or erroneous, the main reason why it took me a long time for writing this response. The authors should try to find a more appropriate title and better names for each section. Furthermore they should reduce the hand-waving explanations in words, and instead use the logic of mathematical formulations. Furthermore, there is a need for improving the language.

We would like to thank the referee for its initial positive assessment and will now discuss each point in detail.

**Special - suggestions for improvements**

1) Throughout the paper change from plural to singular (as already done in Table 2) for: - "sea-ice thicknesses" because there is only one thickness parameter for sea ice.
- "snow thicknesses", even better, use "snow depth".
- "sea-ice concentrations", unless you distinguish between different types of sea ice (e.g. first year, multi year).

We agree. The document has been revised and plural has been changed to singular. (multiple line numbers)

**Special - suggestions for improvements**

2) Section 2 "Data and Methods". This section is poor: `SEP` - formally (longest part of Section 2 without any subsection, followed by short Subsections 2.1 and 2.2, and by a related Section 3)
- logically (models are not well presented, some parameters not defined, equations are missing, Figure 1 confusing, Table 1 inconsistent with text for brine volume fraction), `SEP`
- with respect to the motivation for this paper (e.g. the multilayer model description starts with "The incoherent model used in Maaß et al. (2013) is based on", without any indication why this text is found here, and the same holds for the single-layer model). Start e.g. with "For our analysis we selected ... It is useful because ..."
- and details:

We believe the purpose of this manuscript is the application of the models but not the introduction of them. Therefore, we decided to drastically shorten and rearrange the method section about the radiative transfer models as basically everything is explained in

the referenced papers which explain the models for sea ice purposes in detail. Therefore, we added a motivation, why we exactly used these two radiative transfer models. We changed the arrangement of the subsection, so there so no part without any subsection anymore. We dropped Figure 1 and Table 1 as they were confusing and already covered by the reference papers.

p. 1, line 22: The statement "Microwave radiation is especially useful to derive thin sea ice thicknesses as it is able to penetrate snow and sea ice for more than half a meter". This statement is incorrect because it is far too general. Microwave (1 to 300 GHz) penetration into sea ice is certainly much less than half a meter in most types of sea ice and it is marginal even at the lowest frequency.

Only microwave radiation at 1.4 GHz is investigated in this manuscript. We add this limitation to the revised manuscript to avoid confusion.

"Microwave radiation at 1.4 GHz is especially useful to derive thin sea ice thickness as it is able to penetrate snow and sea ice for more than half a meter and closes the gap to thicker sea ice thickness retrievals of more than 1 meter by using altimetry" [p1,lines 20-24]

p.3, line 6: ORAP5 seems to play an important role. A description would help, a proper reference is a 'must'.

The ORAP5 reanalyses product provides the input data for the radiative transfer models. A reference has been added in this sentence.  [p 3, line 7]

p. 3, lines 14 - 15: What do you mean with "salinity ration"? (misprint?)⌐SEP¬

The sentence has been dropped in the process of reworking the manuscript. Before that, true, it was a spelling error.

p. 5, lines 3-4: Improve this part, but do not try to correct the galactic radiation nor let the atmosphere deviate. What do you mean?: "To account for corrections of the galactic background radiation and atmospheric deviations a simplified atmospheric model (Peng et al., 2013) is taken forced by..." And why do you use a simplified model? What kind of simplification? Use mathematics to show exactly what you did. Readers might want to check.

We used exactly the model describes in Peng et al. 2013 but only called it simplified. To avoid this misunderstanding in the future we dropped the word simplified and only called in the atmospheric model of Peng et al. 2013. [p4, Lines 11-13]

p. 5, line 17: What do you mean with "temperature insulation"? Do you mean "thermal insulation"?

Yes, will be changed in the revised document. [p3, lines 25-26]

p. 5, line 23 once more: "The incoherent model used in Maaß et al. (2013) is based on

radiative transfer equations and describes the emitted radiation from a stratified bare soil". Please be more specific, e.g. by writing "... describes the upwelling brightness temperature at h and v polarisation from bare soil represented by plane-parallel layers with or without surface roughness".

Changed in the revised document with a minor correction as the model of Maaß et al (2013) does not take into account any surface roughness. Therefore, we used the same sentence with "… plane-parallel layers without surface roughness." [p3, Lines 27-29]

p. 5, line 27: The snow density selected seems to me rather high for the usually shallow snow layers found on sea ice. Why do you consider a fixed value? [SEP]

The fixed value has been used as the sensitivity of snow depth on L-Band radiation is very low. Maass et al. 2013 showed in figure 3 that the sensitivity of snow density on brightness temperatures from 260 kg/m^3 up to 340 kg/m^3 is below 1 Kelvin. The value of 330 kg/m^3 has been used as Warren et al. (1999) suggest that value as the climatological average value for March over Arctic sea ice. This evidence has been added to the revised version of the manuscript. [p3, Lines 32-34]

p. 5, line 32 to p. 6, line 3: Modification of the model. Either describe exactly what you did or delete this part. Note that there is a risk of introducing errors.

We decided to rephrase this paragraph and make it shorter, less confusing. Therefore we adapted the whole paragraph.

3) Improve the description of all methods by using appropriate figures for explaining the geometry, angles, etc. as well as mathematical formulas at least for the relevant expressions to enable definitions of the coefficients mentioned (p. 3).

See rearranging of the methods section above. True, but as we only use the radiative transfer models we think there is no need for a comprehensive explanation of the mathematical formulas used as they are all already mentioned in the cited references.

4) p. 7, lines 3-5: Improve physics and timing in "In the melting season, when melt ponds form on sea ice and temperatures begin to rise". Note that temperature rise is much earlier than formation of melt ponds. Explain what you mean and improve the sentence that follows: " SMOS brightness temperatures over sea ice are impossible to connect to a specific sea ice property (Kaleschke et al., 2010)". The logic to the next sentence and its meaning are not clear: "Thus, November and March are the first and the last month, respectively, with full monthly data coverage from SMOS and therefore chosen."

Sentenced rephrased. We need to ensure to have both, cold temperatures and of course no melt ponds on the sea ice. This is the basis for the sea ice thickness assimilation. We now focus in the text on the cold temperatures.

"November and March are the first and the last month in which temperatures are below freezing in the winter season (Vikhamar 2016) and are therefore chosen." [p5, lines 8-11]

5) p. 7, line 15 - : Improve " chosen as values higher than that are not expected to be seen".

We rephrased and added an explanation as the physical maximum of brightness temperatures is capped by 273.15 Kelvin for a surface at 0°C if the emissivity would be at 1.

"Values higher than that are not expected to be seen in the Arctic between November and March as the physical maximum of a surface with temperature at the freezing point would be 273.15 K if the emissivity was 1." [p. 6, lines 1-3]

Also improve the following sentence: " The brightness temperature product consists of vertical and horizontal polarization, which are averaged up to 40 incidence ...". I do not understand. And what do you mean with the sentence that follows? " These brightness temperatures are said to represent L-Band measurements at nadir as brightness temperature changes that are connected to the varying incidence angles are expected to cancel out each other when both polarisations are considered."

The actual goal is not to have a product that represents nadir observations but to have measurements without angle dependency and as big as possible daily data coverage. In case of SMOS L-Band measurements this is achievable by averaging up to 40° incidence angle. We corrected that mistake and improved the sentences in the revised version.

"The brightness temperature product is provided at vertical and horizontal polarization. Although these measurements vary with incidence angles the intensity, defined as the average of horizontally and vertically polarised brightness temperatures, remains almost constant in the range of 0 to 40 degrees over sea ice. By averaging over this incidence angle range we obtain more brightness temperature data per grid point per day reducing considerably the uncertainty." [p. 6, lines 3-8]

6) p. 7 "Sea water correction" What do you mean? Please do not try to correct the water! Please first explain the purpose of this section, and then improve it, especially explain what Figure 2 is supposed to show. Its legend cannot be used to understand what the data clouds mean, nor is the caption of any help. Furthermore the quality of these data is not convincing due to the poor correlation shown. And there must be a reason for the Tb correction. Try to find the error.

We changed the title of the subsection, included it into the methods section and changed the first sentence towards a clearer statement that we apply the bias correction for brightness temperature above sea water only to exclude as many as errors for our brightness temperature comparison of sea ice as we can. This comparison is not supposed to be a main result of this study, but rather to increase the credibility of the brightness temperature comparison above sea ice.

The statement of a difference between simulated and measured brightness temperatures above sea water is a discovery, which we cannot explain by the analysis in this paper, nor can we physically. The error might be due to the representation of wind speed in the model or something totally different. The purpose here is only to have as little influence

on the sea ice brightness temperature simulations as possible.

The caption of figure 2 has been reworked, now hopefully making a clearer statement about the information to see in the figure. [p.8]

7) p. 8 "Brightness temperature comparison". Give a motivation to the reader for not skipping this section. If it is a 'result' section, then please call it accordingly.

We renamed it Result section with subsections called "Brightness temperature comparison" and "Radiative transfer model sensitivity study".

8) p. 10, line 1, also discussion p. 17-18: Explain what the "Kolmogorov-Smirnov-Test " is supposed to check and present the results properly. This is relevant because you use this information for the decision to drop one of the models used. Can you support this decision by physical arguments?

We dropped Figure 6 as it basically shows the same information as Figure 5 does. However, we still use the Kolmogorov-Smirnov-Test which is now briefly introduced in the Discussion section [p.15, line 35]. The result of the test is mainly determined by brightness temperatures over open water, as well as fully saturated brightness temperatures over thick sea ice. There, the MA2013 model overestimates brightness temperatures (already stated in the discussion).

9) p. 13, Figure 5: Nice representation. However the concentration of data points near the two main spots causes problems in the interpretation. It appears that the assessment of thin ice and medium ice concentration is difficult. Think about how to improve the situation, e.g. by omitting some of the data.

Unfortunately, there is no reliable data to make a proper assessment of thin ice and medium sea ice concentration. Therefore, we here decided to concentrate on the open water and 100% sea ice concentration case as it is common in sea ice concentration retrievals (Ivanova et al. 2015). These two cases are our only two reference points which we can use to make a statement about the quality of the radiative transfer models.

As this led to confusion we added another sentence to the discussion section where we make the statement about the favorable model. [p 15. Lines 13-15]

10) p. 13, lines 16-17: "we observe an underestimation of sea ice concentration" underestimation by what, i.e. which model?

By the reanalysis, has been changed in the revised version.

"Following, we observe an underestimation of sea ice concentration and an overestimation of sea ice thickness in the reanalysis over a range of two weeks, whereas the SMOS brightness temperature only deviate more than 20 K for 5 days." [p. 14, lines 9-10]

11) p. 15, Figure 8: Exchange the two legends in order to be close to the respective y

axis, and clarify 'growth model' with respect to caption (Lebedev ?). What is the role "Lebedev" is playing here (missing in Sect 2).

A related subsection about the sea ice growth model from Lebedev has been added to section 2. The legend in Figure 8 has been exchanged.

10   In this study, we investigate the Arctic-wide performance of the radiative transfer models of Kaleschke et al. (2010) and Maaß et al. (2013) to account for diverse atmospheric and oceanic conditions and to identify the most important input parameters for a sea ice thickness application. In preparation for a brightness temperature assimilation, we concentrate on the input data of the global ocean reanalysis product ORAP5 (Ocean ReAnalysis Pilot 5) produced by the ECMWF (Zuo et al., 2015). We evaluate which radiative transfer model to use for assimilating sea ice thickness into the ORAP5  reanalysis by

15   comparing simulated and observed brightness temperatures from the radiative transfer models with ORAP5 input data and SMOS observations, respectively.

**2   Data and Methods**

**2.1   Radiative transfer models**

20   For our analysis we selected the radiative transfer models ~~provide brightness temperatures as a function of temperature, snow- and sea ice thickness, incidence angle and the permittivity (Fig. ??). The latter is calculated with the snow, sea ice and sea water temperatures (as well as the snow/sea ice interface temperature), the bulk sea ice and water salinities and the snow and sea ice thicknesses. The sea ice salinity is estimated with the empirical approach of Ryvlin (1974) making the ice salinity a function of the sea surface salinity, the sea ice thickness , the salinity ration of the bulk ice salinity at the end~~

25   ~~of the sea ice growing season and the growth rate coefficient. The latter two are taken from Kovacs (1996) who derived a value of 0.175 for the ice salinity ration from observational data in the Arctic and 0.5 for the growth rate coefficient, as also suggested by Ryvlin (1974). To obtain the ice/snowinterface we calculate the thermal conductivity of ice with the snow surface temperature and ice salinity (Untersteiner, 1964). Using this, we are able to determine the bulk sea ice temperature for our sea ice slab by assuming the heat fluxes are in equilibrium (Maykut and Untersteiner, 1971). We calculate the brine salinity~~

30    of Kaleschke et al. (2010) and Maaß et al. (2013) to simulate brightness temperatures above sea water and ice at 1.4 GHz. The two models have been chosen because the model of Kaleschke et al. (2010) is a rather simple single-layer

model that has been successfully applied for operational sea ice thickness retrieval (Kaleschke et al., 2012), whereas the model of Maaß et al. (2013) consists of multiple layers and is yet only used for sensitivity studies above snow. Both models provide brightness temperatures as a function of ~~bulk ice temperature (Pounder, 1965). As the polynomial coefficients for the brine salinity calculation are only provided for 1 and 2 GHz, we linearly interpolate between these two frequencies to determine the coefficients for 1.4 GHz. By taking the ice and brine density, as well as the bulk ice temperature, we are able to calculate brine volume fraction using equations valid for ice temperatures below -2°C from Cox and Weeks (1983) and above -2°C from Leppäranta and Manninen (1988). Finally, the permittivity of sea ice is derived by an empirical relationship to the brine volume fraction (Vant et al., 1978) (for a summary of references see table ??)~~ the considered layers' temperature, thickness and permittivity.

~~In order to represent sea ice brightness temperature measurements in partially covered data points over the open ocean, the models linearly include sea ice fractional coverage in the calculations for each grid cell. Additionally, both models consider the subpixel-scale heterogeneity of sea ice thicknesses with a statistical ice thickness distribution. We calculate the brightness temperatures for ten linearly divided sea ice thickness bins with a maximum of 2 meter thickness. Then, we translate the mean sea ice thickness from the input data to a sea ice thickness distribution derived by observational data (As used in Algorithm II* by Tian-Kunze et al. (2014)). The final brightness temperature is the average of the ten respective bins weighted by the sea ice thickness distribution.~~

~~Applied methods to obtain auxiliary parameters for brightness temperatures calculations above sea ice and snow. The numbers in the first row refer to the numbers in Fig.??. No. Parameter References 1 Bulk ice salinity Ryvlin (1974) 2 Ice thermal conductivity Untersteiner (1964) 3 Snow/Ice interface and bulk temperatures Maykut and Untersteiner (1971) 4 Brine salinity Vant et al. (1978); Leppäranta and Manninen (1988) 5 Brine volume fraction Cox and Weeks (1983) 6 Snow permittivity Tiuri (1984) 7 Sea ice permittivity Vant et al. (1978) 8 Brightness temperatures Kaleschke et al. (2010), Maaß et al. (2013)~~

[revised manuscript text omitted]

An overview of the influence of single parameters on the brightness temperature simulation using KA2010 and MA2013 is given in table ??. One parameter varies over a range of values, whereas all other parameters are fixed to a default value similar to the method used in Fig. 5. The range of values is set to be the 99% quantile of its largest simulated change within the month of November. This calculation is performed twice, once for a thin sea ice scenario and once for a thick sea ice

10 scenario. As expected it is shown that the sea ice concentration and sea ice thickness has the greatest influence on the brightness temperature calculation on a range from 0 - 100% and 0-0.77m. The brightness temperature difference is up to 149 Kelvin in the case of thicker sea ice and the MA2013 model. The effect is less strong in KA2010 but still dominating. However, the sea ice temperature, snow thickness and sea ice salinity are also highly important. A change of snow thickness from no snow to a 40 cm snow layer will change the brightness temperature around 28 Kelvin in both models for thin and thick sea ice. Third

15 most important is the sea ice temperature accounting for brightness temperature changes up to 24 Kelvin in a range from -35 to -5 °C, slightly more in the KA2010 model. The sea ice salinity accounts for around 20 Kelvin in case of thin sea ice and 12 or 6 Kelvin for KA2010 or MA2013, respectively, in case of thick sea ice. The influence of sea ice salinity appears to be more sensitive to sea ice thickness in MA2013 than in KA2010, as we see a larger change between thin and thick sea ice. For the sea surface salinity and the snow density, we are do not observe a great influence. These above examined values may alter

20 significantely by changing the default value, but will give us good insight into the relation of physical parameters to simulated brightness temperatures.

An overview of the influence of single parameters on the brightness temperature simulations using KA2010 and MA2013 is given in table ??. One parameter varies over a range of values, whereas all other parameters are fixed to a default value similar to the method used in Fig. 5. The range of values is set to be the 99% quantile of its largest simulated change of the value

25 within the month of November. At that time, changes are assumed to be greatest as it is in the middle of the sea ice growth season. This calculation is performed twice, once for thin sea ice (10cm) and once for a thick sea ice (50cm) except in case of varying sea ice thickness. As expected it is shown that the sea ice concentration and sea ice thickness has the largest influence on the brightness temperature calculation on a range from 0 - 100% and 0 - 0.77 m, respectively. In case of thin sea ice the effect on brightness temperatures appears to be even more sensitive for sea ice thickness changes up to 159 K in the MA2013

30 model. However, a monthly change of sea ice temperature, snow thickness and sea ice salinity also has a noticable impact on the brightness temperature calculation. The sea ice temperature is most important with changes up to 14 Kelvin in the KA2010 model. The snow thickness and sea ice salinity are roughly half of that. By comparing the impact with the thick sea ice scenario the influence of the sea ice temperature, snow thickness and sea ice salinity decreases, whereas sea ice concentration becomes more dominant. This is especially true for the model of MA2013, in which the sea ice concentration and the sea ice thickness

35 influence is greatest, most pronounced in the thin sea ice scenario. These above examined values may alter significantely by changing the default value, which is here chosen to represent typical Arctic and Antarctic first year ice. Still, it will give us insight into the relation of physical parameters to simulated brightness temperatures.

[revised manuscript text omitted]
. For a brightness temperature assimilation this would be rather detrimental, as we only expect benefits from SMOS-based ice thickness assimilation thin sea ice regions because microwave radiation is not able to distinguish between thicker sea ice. In contrast, KA2010 shows good agreement in the central Arctic area. For brightness temperature assimilation~~

30 ~~purposes that would be clearly beneficial as the main assimilation should take place in regions with thin sea ice rather than in the central Arctic. Additionally, MA2013 shows a rapid brightness temperature rise in the transition from open water towards the very first centimeters of sea ice (Kaleschke et al. (2010), Maaß et al. (2013)). 
[revised manuscript text omitted]

Tiuri, M. E.: The Complex Dielectric Constant of Snow at Microwave, IEEE Journal of Oceanic Engineering, 9, 377–382, 1984.

Ulaby, F. T., Moore, R. K., and Fung, A. K.: Microwave remote sensing.Active and passive., Kansas Univ.; Lawrence, KS, United States, 1981.

30  Untersteiner, N.: Calculationsof TemperatureRegime and Heat Budget of Sea Ice in ,the Central Arctic, 69, 4755–4766, 1964.

Vant, M. R., Ramseier, R. O., and Makios, M.: The complex-dielectric constant of sea ice at frequencies in the range 0.1-40 GHz., Journal of Applied Physics, 49, 1264–1280, 1978.

Vikhamar-Schuler, D., Isaksen, K., Haugen, J. E., Tommervik, H., Luks, B., Vikhamar-Schuler, T., and Bjerke, J. W.: Changes in Winter Warming Events in the Nordic Arctic Region, Journal of Climate, 29, 6223 – 6244, doi:10.1175/JCLI-D-15-0763.14, 2016.

35  Warren G., S., Rigor G., I., Untersteiner, N., Radionov F., V., Bryazgin N., N., Aleksandrov I., Y., and Colony, R.: Snow Depth on Arctic Sea Ice, Journal of Climate, 1999.

Xie, J., Counillon, F., Bertino, L., Tian-kunze, X., and Kaleschke, L.: Benefits of assimilating thin sea ice thickness from SMOS into the TOPAZ system, The Cryosphere, 10, 2745–2761, doi:10.5194/tc-10-2745-2016, 2016.

Yu, Y. and Lindsay, R. W.: Comparison of thin ice thickness distributions derived from RADARSAT Geophysical Processor System and advanced very high resolution radiometer data sets, Journal of Geophysical Research, 108, 3387, doi:10.1029/2002JC001319, 2003.

Yu, Y. and Rothrock, D. a.: Thin ice thickness from satellite thermal imagery, Journal of Geophysical Research, 101, 25 753,

5  doi:10.1029/96JC02242, 1996.

Zhang, J. and Rothrock, D. a.: Modeling Global Sea Ice with a Thickness and Enthalpy Distribution Model in Generalized Curvilinear

685    Coordinates, Monthly Weather Review, 131, 845–861, doi:10.1175/1520-0493(2003)131, 2003.

Zuo, H., Balmaseda, M. a., Boisseson, E., and Hirahara, S.: A New Ensemble Generation Scheme for Ocean Analysis, ECMWF Tech Memo, 795.

Zuo, H., Balmaseda, A. M., and Mogensen, K.: The new eddy-permitting ORAP5 ocean reanalysis: description, evaluation and uncertainties in climate signals, Climate Dynamics, pp. 1–21, doi:10.1007/s00382-015-2675-1, 2015.

---

## Author Response (AR2)

Comments to the Author:
I thank the authors for their efforts to revise the manuscript. Before publishing I have a few remaining edits I suggest the authors make:

We would like to thank the referee for its initial positive assessment and will now discuss each point in detail.

Line 12, page 2, you mention ice salinity twice. After that sentence I find it confusing to first state the RT model Kaleschke et al. 2010 used and then say a second RT model was utilized, when in fact this model of Maas et al. (2013) was not used for thickness but rather for snow depth. I suggest to better streamline this discussion.

The repetition of the word salinity has been removed in the revised version.

We believe many misunderstandings were reasoned in the structuring of the introduction. Therefore we restructured the introduction with the focus on the assimilation of sea ice parameters and brightness temperatures instead.

Line 25, page 2. I don't see the value of this sentence and I would remove it.

Has been removed in the revised version.

You are just mentioning how age has been used as a proxy for thickness in PIOMAS, it's not really relevant how it compares to observations as that is not the point of this paper. In addition, I think you could also streamline this discussion to simply refer to the fact that SIC has been assimilated, but SIT has mostly not been. I don't quite follow the reasoning why inconsistency between SIC of the model and SIC based on SMOS is a limitation for SIT assimilation. This entire paragraph needs to be streamlined and made more clear to get the main message across. I don't feel a strong case has been shown that in order to improve SIT assimilation you need to improve SIC and surface temperature. It seems to me you are arguing that's the reason why SIT is not assimilated, when in reality it could be because the data are not accurate enough yet for assimilation. The introduction seems to continually switch back in forth between discussing assimilating Tbs or SIT. For example on page 3, paragraph beginning on line 10 you state in preparing for a TB assimilation you are trying to figure out which RT model to use for assimilation SIT.
I agree that assimilation of Tbs is likely preferable to the derived fields but this case needs to be better stated and supported in the introduction. It remains a bit confusing the way the introduction is currently written.

This has been addressed in the new order of the introduction. We now start with the assimilation of sea ice parameters and end with the assimilation of brightness temperatures.

Methods: I think it would be better to lead off with the SMOS data since the aim is to assimilate SMOS Tbs.

The methods section now starts with SMOS data.

You should also provide a clear reason why the two RT models were chosen. One seems suited for relating SMOS Tbs to thin ice thickness and the other for snow depth over thicker ice, but what is the main objective then here? The conclusion that the KA model is better for thickness seems already known. If the objective is for SIT assimilation then I'm curious why the Maas model was evaluated in the first place despite being of interest to show how the two RT models are sensitive to different input parameters. My question is, has this already been done in previous studies, or is this the first study to do that? I think in this section you could benefit from a schematic that shows all the processing steps and the data input/output. This could help greatly in understanding exactly why this study is being conducted.

We believe this comment should be much clearer with the new structured introduction. Both models are utilized to simulate brightness temperatures above sea ice. However, the more complex one (maas model) takes more parameters into account. This is no benefit yet as we do not provide all of these parameters but it can be of big advantage in future research. There is no paper of our knowledge that examined the sensitivity of input parameters of these radiative transfer models.

What is the reason for the 3K bias over the open ocean? If part of it is from land contamination, why not expand the land mask out a few more pixels to remove any land influence? Do you expect the offset to remain linear (i.e. a constant 3.1K bias?). Figure 1 does not show a great fit even with the bias added, so some comment on the large range of the observed Tbs is needed.

The variation of SMOS brightness temperatures above open water is much larger than these we can simulate with radiative transfer models. The reason is the simplicity of the model. We are e.g. not able to take the wind speeds or white caps into account that will increase the observed brightness temperature. However, the bias is just an observation during the evaluation process that needs to be filtered out as good as we can but in fact, we do not know the exact reason why the observed brightness temperature is higher than the simulated ones. A comment has been added to the methods section.

Page 11, last paragraph seems to have something missing with the sentence starting with "In is area..."

Corrected.

[revised manuscript text omitted]

---

## Author Response (AR3)

UHH · CEN Institut für Meereskunde
Bundesstraße 53 · 20146 Hamburg

Prof. Dr. Julienne Stroeve
Editor of The Cryosphere

**Prof. Dr. Lars Kaleschke**
Universität Hamburg
CEN Centrum für
Erdsystemforschung
und Nachhaltigkeit
Institut für Meereskunde
Bundesstraße 53
20146 Hamburg

Tel.   +49   (0)40   42838-6518
Fax   +49   (0)40   42838-7471
lars.kaleschke@cen.uni-hamburg.de
www.ifm.cen.uni-hamburg.de
www.cen.uni-hamburg.de

Datum     2. Februar 2018

AZ

**Minor revision of tc-2016-273     Submitted on 25 Nov 2016**
**Arctic sea ice signatures: L-Band brightness temperature sensitivity comparison using two**
**radiation transfer models by Friedrich Richter et al.**

Dear editor,

thank you for your allowance of extra time to do the necessary revision of the paper. We have largely re-written, restructured and streamlined the introduction, discussion, summary and conclusion. It was indeed wrong that ice age from NSIDC is used for assimilation and the corresponding reference was incorrect.

Please find attached a diff version of the text with changes to version 4 highlighted in colors. Unfortunately, the automated latex diff did not highlight the changes in section titles and further references. These are the not-highlighted changes:

We changed „Summary and outlook" to „Summary and conclusion"

We added the following new references: Chen et al. (2017), Ricker et al. (2017), Sakov et al. (2012), Simmons et al. (2016), Zhuo et al. (2017)

Thank you for your effort!

Best regards,

Hamburg, 2. Februar 2018          Lars Kaleschke

**Editor Decision: Publish subject to minor revisions (Editor review)** (24 Sep 2017) by Julienne Stroeve
Comments to the Author:
I thank the authors for attention to the additional revisions. However the first paragraph of the manuscript is still not quite right. The first sentence is incomplete, and it is not true that ice age from NSIDC is used as a proxy for ice thickness in PIOMAS. Can you say which forecasting models assimilate ice age?

In general, I still feel the introduction could be better written and streamlined. Many sentences do not flow well together. This is likely a reflection of writing style, but several grammatical errors remain. I suggest the authors more carefully proofread the manuscript.

Finally, I still find the discussion somewhat confusing, you state in the introduction: 
[revised manuscript text omitted]